# OCEAN: Offline Chain-of-thought Evaluation and Alignment in Large Language Models

**Junda Wu**[1][*], **Xintong Li**[1][*], **Ruoyu Wang**[2], **Yu Xia**[1], **Yuxin Xiong**[1], **Jianing Wang**[3],
**Tong Yu**[4], **Xiang Chen**[4], **Branislav Kveton**[4], **Lina Yao**[2,5], **Jingbo Shang**[1], **Julian McAuley**[1]
[1]UC San Diego    [2]The University of New South Wales
[3]East China Normal University    [4]Adobe Research    [5]CSIRO's Data61
`{juw069,xil240,yux078,y7xiong,jshang,jmcauley}@ucsd.edu`
`{ruoyu.wang5}@unsw.edu.au  lygwjn@gmail.com`
`{tyu,xiangche,kveton}@adobe.com  lina.yao@data61.csiro.au`

## Abstract

Offline evaluation of LLMs is crucial in understanding their capacities, though current methods remain underexplored in existing research. In this work, we focus on the offline evaluation of the chain-of-thought capabilities and show how to optimize LLMs based on the proposed evaluation method. To enable offline feedback with rich knowledge and reasoning paths, we use knowledge graphs (KGs) (*e.g.*, Wikidata5M) to provide feedback on the generated chain of thoughts. Due to the heterogeneity between LLM reasoning and KG structures, direct interaction and feedback from knowledge graphs on LLM behavior are challenging, as they require accurate entity linking and grounding of LLM-generated chains of thought in the KG. To address the above challenge, we propose an offline chain-of-thought evaluation framework, OCEAN, which models chain-of-thought reasoning in LLMs as a Markov Decision Process (MDP), and evaluate the policy's alignment with KG preference modeling. To overcome the reasoning heterogeneity and grounding problems, we leverage on-policy KG exploration and reinforcement learning to model a KG policy that generates token-level likelihood distributions for LLM-generated chain-of-thought reasoning paths, simulating KG reasoning preference. Then we incorporate the knowledge-graph feedback on the validity and alignment of the generated reasoning paths into *inverse propensity scores* and propose KG-IPS estimator. Theoretically, we prove the unbiasedness of the proposed KG-IPS estimator and provide a lower bound on its variance. With the off-policy evaluated value function, we can directly enable off-policy optimization to further enhance chain-of-thought alignment. Our empirical study shows that OCEAN can be efficiently optimized for generating chain-of-thought reasoning paths with higher estimated values without affecting LLMs' general abilities in downstream tasks or their internal knowledge.

## 1 Introduction

Offline policy evaluation aims to estimate a target policy model's performance with only collected data, without requiring direct interactions between the target policy and realistic environments. Previous offline evaluation methods focus on decision-making policies in recommender systems (Li et al., 2011), healthcare (Bang & Robins, 2005), and other scenarios where online experimentation is costly (Thomas et al., 2015; Bhargava et al., 2024), risky, and impractical (Yu et al., 2021). Recent studies in LLMs leverage human feedback to align models' behaviors with human preferences in single-turn generation (Ouyang et al., 2022; Rafailov et al., 2024) and multi-step reasoning tasks (Joshi et al., 2024). In addition, complicated LLM agentic frameworks, involving multi-agent collaboration, orchestration, and cooperation, rely heavily on efficient (Roucher et al., 2025; Wu et al., 2023a), robust (Masterman et al., 2024; Nguyen et al., 2024a), and proactive (Yao et al., 2023; Xia et al., 2025; Ma et al., 2023) chain-of-thought reasoning abilities, which need to be finetuned offline

---

[*]These authors contributed equally to this work.

(Putta et al., 2024) before deploying them online. Due to the high cost of deploying LLMs online and interacting with human feedback, Bhargava et al. (2024) further enables offline evaluation of LLMs from logged human feedback to align LLMs' response generation.

However, considering annotators may not have comprehensive knowledge in various types of knowledge backgrounds, human feedback on chain-of-thought reasoning (Joshi et al., 2024) can be more challenging to collect. In addition, since chain-of-thought reasoning involves a sequential decision-making process, the volume of collected human feedback may increase exponentially. Due to such challenges, conventional reinforcement learning from human feedback (RLHF) methods (Ouyang et al., 2022; Bai et al., 2022a) can suffer from training inefficiencies and scalability issues.

Motivated by recent works in using knowledge graphs (KGs) as side information for prompt engineering (Wang et al., 2024c; Xia et al., 2024b), self-correction (Zhao et al., 2023; Wang et al., 2023; Li et al., 2024b; Wu et al., 2024b), evaluating chain-of-thought (Nguyen et al., 2024b), and model fine-tuning (Wang et al., 2024b; Tang et al., 2024), we propose leveraging KGs as weak yet controllable knowledge reasoners to effectively measure the alignment between LLMs' multi-step chain-of-thought reasoning and multi-hop KG trajectories by *inverse propensity scores* (IPS) (Joachims et al., 2017). Unlike the chain-of-thought evaluation method (Nguyen et al., 2024b), which depends on accurate chain-of-thought grounding in specific KGs, we propose to verbalize KG trajectories and develop a KG policy as a verbal reasoning mechanism over the graphs. This approach bridges the gap between KG and LLM reasoning and generalizes the KG policy to various LLMs.

To enable controllable chain-of-thought alignment in LLMs, we principally track LLMs' decision-making process in generating chain-of-thought reasoning steps, by formulating the process as a Markov Decision Process (MDP) whose goal is to reach the correct final answer with minimal knowledge exploration and exploitation Lissandrini et al. (2020b;a); Wu et al. (2024a). Then, we propose offline chain-of-thought evaluation and alignment, OCEAN, which evaluates the generated chain of thoughts from off-policy LLMs through collected offline data samples with feedback from a KG. The improved Knowledge Graph - Inverse Propensity Scores (KG-IPS) approach considers the effects of feedback from the KG policy that aligns the model's chain-of-thought generation and the behavior policy, which prevents model degeneration. We prove that the KG-IPS estimator provides an unbiased estimate of the target policy, with a lower bound for the variance, and establish confidence intervals using sub-Gaussian concentration inequalities. To enable direct optimization of LLM policies, we leverage the proposed KG-IPS policy evaluation approach for LLM fine-tuning by directly maximizing estimated policy values through gradient descent. Then we empirically evaluate the optimized LLM policy on three types of chain-of-thought reasoning tasks, and demonstrate the effectiveness of the proposed policy optimization method, without affecting LLMs' generalizability or generation quality. We summarize our contributions as follows:

- We propose an offline evaluation framework, OCEAN, which bridges the heterogeneity between LLM and KG reasoning, for effective evaluations of chain-of-thought.
- With the evaluation framework, we further develop a direct policy optimization method which enables efficient alignment with automatic feedback from the KG.
- To facilitate the evaluation and optimization, we model the KG preference and derive feedback by developing a policy which verbalizes KG trajectories.
- We provide a theoretical analysis of the unbiasedness and establish a lower bound for the variance of our KG-IPS estimator.
- Through comprehensive experiments, we demonstrate OCEAN's effectiveness in aligning LLMs' chain-of-thought reasoning through direct optimization of the estimated policy value. OCEAN also achieves better performance on various downstream tasks without affecting LLMs' generalizability.

## 2 RELATED WORK

**Offline Policy Evaluation** Offline policy evaluation (OPE) is essential when online policy learning is risky and impractical (Levine et al., 2020). OPE has been applied to various practical applications, including evaluating the recommender system's behavior with offline collected user feedback (Gilotte et al., 2018; Jeunen, 2019). Recent work (Gao et al., 2024) also develops an OPE estimator for LLM evaluation based on human feedback. Different from previous works, we study and

formulate chain-of-thought generation in LLM as an MDP and use knowledge graph reasoning as automatic feedback to develop a KG-IPS policy value estimator.

**LLM Alignment** Reinforcement Learning from Human Feedback (RLHF) has been the dominant approach, optimizing LLMs using human-annotated data to align model behavior with user preferences (Ouyang et al., 2022; Bai et al., 2022a). DPO (Rafailov et al., 2024) and RRHF (Yuan et al., 2023) are proposed to reduce the training instability of RLHF. Wu et al. (2023b) utilizes varying densities of human feedback to offer fine-grained rewards for RL finetuning, and Sun et al. (2024a) focuses on aligning LLMs with reward models driven by human-defined principles. To address RLHF's limitations such as heavy reliance on human input, alternative approaches like Reinforcement Learning from AI Feedback (RLAIF) (Bai et al., 2022b; Lee et al., 2023; Liu et al., 2023) and self-alignment methods (Sun et al., 2024b) have been proposed, using AI-generated feedback to scale and automate alignment. Despite advancements, a key challenge remains in aligning LLMs' internal knowledge with their reasoning, resulting in flawed reasoning even after factual errors are corrected. Our approach focuses on improving chain-of-thought alignment by modeling reasoning paths as an MDP and using KGs to ensure both factual accuracy and human-like reasoning.

**Chain-of-thought Reasoning** Chain-of-thought prompting has been widely applied to elicit the strong reasoning abilities of LLMs (Wei et al., 2022; Chu et al., 2023; Xia et al., 2024a). By decomposing a complex problem into a sequence of intermediate sub-tasks, LLMs can focus on important details and solve the problem step by step (Huang & Chang, 2023; Yu et al., 2023). Despite the remarkable performance improvements, recent studies have found that LLMs often generate unfaithful chain-of-thought reasoning paths that contain factually incorrect rationales (Turpin et al., 2023; Lanham et al., 2023). To address this, a number of works leverage LLMs' self-evaluation abilities to verify and refine each reasoning step (Ling et al., 2023; Madaan et al., 2023). As the factual errors in the generated chain-of-thought may also be caused by the limited or outdated parametric knowledge of LLMs, recent methods incorporate external knowledge sources to further edit unfaithful content in the reasoning path (Zhao et al., 2023; Wang et al., 2023; Li et al., 2024b; Wang et al., 2024d;a). While these methods focus on knowledge augmentation and editing through prompts, our method, in comparison, directly aligns LLM internal knowledge with faithful and factual chain-of-thought, which avoids potential knowledge conflicts between parametric and non-parametric knowledge when generating reasoning paths.

## 3 PRELIMINARY

We first provide the formulation of chain-of-thought reasoning in LLMs as an MDP. Then we discuss conventional knowledge graph reasoning, as an alternative to free-form generation by verbalizing structured knowledge graph reasoning paths into natural language, which is more statistically controllable and generates faithful reasoning paths to the knowledge graph.

### 3.1 PROBLEM FORMULATION: CHAIN-OF-THOUGHT AS AN MDP

Given the prompt instruction $q$, chain-of-thought reasoning process in a causal language model $\pi_\theta$ includes the generation of a trajectory of reasoning steps $\boldsymbol{c} = (c_1, c_2, \ldots, c_T)$, before the final answer prediction $y$,

$$c_t \sim \pi_\theta(\cdot \mid q, c_{<t}) \quad c_{<t} = (c_1, \ldots, c_{t-1}), \quad y \sim \pi_\theta(\cdot|q) = \pi_\theta(y|q, \boldsymbol{c}) \prod_{t=1}^{T} \pi_\theta(c_t|q, c_{<t}),$$

where each reasoning step $c_t$ comprises a sequence of tokens and the number of reasoning step $T$ is determined by the model's generation. Controllable chain-of-thought generation can be challenging due to its nature in autoregressive sequential sampling (Lin et al., 2020), which produces a high-dimensional action space in sampling a reasoning step $\pi_\theta(c_t|q)$ containing multiple tokens.

Chain-of-thought reasoning can be viewed as a Markov Decision Process (MDP) (Sutton, 2018): starting with the instruction prompt $q$, the LLM sequentially decides and generates the next-step reasoning path $c_t$ that navigates until it arrives at a target final answer $y$. Given the LLM policy $\pi_\theta$, at time step $t$, each **state** $s_t \in \mathcal{S}$ comprises of the instruction prompt $q$ and previously generated reasoning paths $(c_i)_{i=0}^{t-1}$. The **action** space $\{1, \ldots, |\mathcal{V}|\}^{N_t}$ in LLMs is a sequence of $N_t$ tokens as

a knowledge graph entity or relation identified on a single thought, sampled from an identical and finite vocabulary set $\mathcal{V}$. The LLM policy $\pi_\theta$ samples next-step thought based on current state as $a_t \sim \pi_\theta(\cdot \mid s_t)$, which is a sub-sequence in the reasoning path $a_t \subseteq c_t$ identified on the knowledge graph. The surrounding context $c_t \setminus a_t$ other than the knowledge graph entity or relation is deterministically generated by LLMs. The **transition** in chain-of-thought is concatenating each reasoning path to the current state as $s_{t+1} = [s_t, c_t]$. Then the **reward** function is to evaluate each thought given the state as $r_t = r(s_t, c_t)$. Although such formulation of chain-of-thought enables direct LLM on-policy optimization via reinforcement learning, direct interaction with knowledge graphs to collect per-step reward in LLMs can be practically challenging and require a large effort of engineering due to the discrepancy between the unstructured generation of LLMs and structured knowledge graphs (Pan et al., 2024). Therefore, we propose to offline evaluate and optimize the target policy aligning with knowledge graph preference.

## 3.2 VERBALIZED KNOWLEDGE GRAPH REASONING

In contrast to chain-of-thought reasoning, conventional knowledge graph reasoning methods (Lin et al., 2018; Saxena et al., 2020) sample a entity-relation pair $(r_t, e_t)$ at step $t$ from a subset of the graph $\mathcal{G} = (\mathcal{E}, \mathcal{V})$ consisting of the outgoing edges of current entity $e_{t-1}$,

$$(r_t, e_t) \in \{(r', e')|(e_{t-1}, r', e') \in \mathcal{G}\}, \tag{1}$$

where the transition feasibility of the entity $e_{t-1}$ to all the outgoing edges is entirely determined by $\mathcal{G}$. Knowledge graph reasoning starts with a triplet $(e_0, r_1, e_1)$ and produces a chain of triplets $\boldsymbol{h} = (e_0, r_1, e_1, \ldots, r_T, e_T)$ by sampling from a policy $\mu$,

$$(r_t, e_t) \sim \mu\left((r_t, e_t)|e_0, r_1, e_1, \ldots, r_{t-1}, e_{t-1}\right), \tag{2}$$

where the goal of such knowledge graph exploration is to arrive at the correct answer entity at the end of the search step $T$. By knowledge graph exploration, we can collect a set of trajectories $\mathbb{H} = \{\boldsymbol{h}_k\}_{k=1}^K$, which are used to estimate a parametric probabilistic policy $\mu_\phi$ as a proxy to model the preference of the knowledge graph.

To align the action space between the knowledge graph preference policy $\mu_\phi$ and the target policy $\pi_\theta$, we leverage a small language model as the backbone of $\mu_\phi$ and fine-tune the model on verbalized trajectories as natural language contexts. Inspired by existing efforts in verbalizing structured knowledge graphs into natural language query (Seyler et al., 2017) and context (Agarwal et al., 2020; Wang et al., 2022a), we leverage the GPT-4 (Achiam et al., 2023) model $f$ to verbalize each chain of triplets $\boldsymbol{h}$ into a chain-of-thoughts $\boldsymbol{c} = f(\boldsymbol{h})$. The verbalized knowledge-graph trajectories are used to model knowledge graph preference in Section 4.2.

## 4 OCEAN: OFFLINE CHAIN-OF-THOUGHT EVALUATION AND ALIGNMENT

We propose an offline evaluation of the chain-of-thought generation process aligned with knowledge graph preference. The off-policy estimator can be used for policy optimization that aligns LLMs with more faithful reasoning paths from knowledge graphs (Lin et al., 2023). We develop a small language model as a behavior policy that models the knowledge graph preference. In Figure 1, we illustrate the workflow of our proposed framework OCEAN.

## 4.1 OFFLINE EVALUATION AND OPTIMIZATION

One of the most broadly used offline evaluation approaches is *inverse propensity scores* (Ionides, 2008; Dudík et al., 2011), which has been used for LLM-based offline policy evaluation for various purposes (Bhargava et al., 2024; Dhawan et al., 2024; Wu et al., 2022). Given the offline logged chain-of-thought trajectories $\mathcal{D} = \{\tau_i\}_{i=1}^N$, where $\tau_i = (s_t^{(i)}, c_t^{(i)}, r_t^{(i)}, s_{t+1}^{(i)})_{t=0}^{T_i}$, we propose a KG-IPS estimator considering two-folded weights of entity tokens in the knowledge graph preference policy $\mu_\phi$ and of non-entity tokens in the base LLM policy $\pi_0$ ,

$$\hat{V}_{KG-IPS}(\theta) = \frac{1}{N}\sum_{i=1}^N \frac{1}{T_i}\sum_{t=1}^{T_i} \frac{1}{|c_t^{(i)}|}\sum_{v \in c_t^{(i)}} \frac{\pi_\theta(v|s_t^{(i)})}{\lambda(v|s_t^{(i)})} \log \pi_0(v|s_t^{(i)}), \tag{3}$$

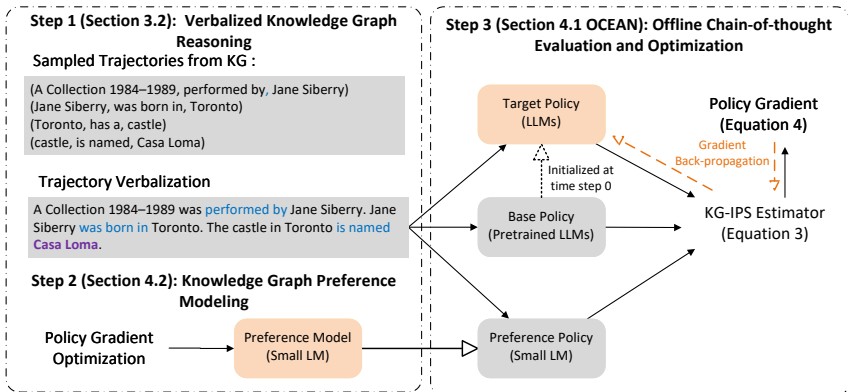

Figure 1: An illustration of our framework OCEAN. We illustrate the framework in three steps. **Step 1** samples trajectories from the Wikidata5M knowledge graph and verbalizes the reasoning trajectories. With the collected trajectories, **Step 2** trains a knowledge graph preference modeling, which is fixed and used during chain-of-thought evaluation and optimization in **Step 3** We highlight the learnable models in red and the frozen models in gray.

where $T_i$ is the CoT steps for trajectory $i$; $|c_t^{(i)}|$ is the number of tokens within the reasoning step $c_t^{(i)}$. The propensity ratio in equation 3 is $\pi_\theta(v|s_t^{(i)})/\lambda(v|s_t^{(i)})$, where $\lambda(v|s_t^{(i)}) = \mathbf{1}\{v \in a_t^{(i)}\} \cdot \mu_\phi(v|s_t^{(i)}) + \mathbf{1}\{v \in c_t^{(i)} \backslash a_t^{(i)}\} \cdot \pi_0(v|s^{(i)})$. In this formulation, the two probabilities are combined to account for cases where $\mu_\phi(v \mid s_t^{(i)})$ might be undefined if those tokens are surrounding texts that cannot be identified on the knowledge graph. Specifically, we replace the undefined probability with the fallback probability $\pi_0(v \mid s^{(i)})$. We follow (Zhang et al., 2024) to use the log-likelihood score of each token in the base policy $\pi_0$ as the reward function. Establishing the unbiasedness of the KG-IPS estimator is essential for reliable policy evaluation (Jiang & Li, 2016; Bhargava et al., 2024). We formalize this in the following lemma:

**Lemma 1.** *The KG-IPS estimator provides an unbiased estimate of the target policy $\pi_\theta$.*

Intuitively, by re-weighting the token-level likelihoods from the target policy $\pi_\theta$ with the behavior policy $\mu_\phi$ (for entity tokens) and the base policy $\pi_0$ (for non-entity tokens), we ensure that our estimator compensates for the off-policy distribution, leading to an unbiased estimate of the true value function. The detailed proof is provided in Appendix A.

The standard IPS estimator is known to have a high variance (Metelli et al., 2018) considering large behavior discrepancies ($\pi_\theta(v|s_t^{(i)})/\mu_\phi(v|s_t^{(i)})$) between the behavior policy $\mu_\phi$ and the target policy $\pi_\theta$. In addition, by separately weighting the entity and non-entity tokens with $\mu_\phi$ and $\pi_0$ respectively, we avoid the increasing variance accumulated from the long chain-of-thought reasoning process and maintain the LLM's behaviors on non-entity tokens without model degeneration. To further formalize our approach and illustrate the variance inherent in the KG-IPS estimator, we present the following Lemma, which provides a lower bound on the variance,

**Lemma 2.** *The variance of the KG-IPS estimator is lower bounded by $\Omega(\frac{M^2}{n})$, where $M$ denotes the maximum value of the weighted terms, and $n$ is the number of samples. For a target policy $\pi_\theta$, let the true value function be defined as $V(\theta) := \mathbb{E}\left[\frac{\pi_\theta(e|s_t)}{\mu_\phi(e|s_t)}r_t\right]$, where $r_t \in [0,1]$ is the reward associated with selecting entity $e$ in state $s_t$ and $\mu_0$ is the behavior policy under which the data is collected. Applying the concentration inequality for sub-Gaussian variables, the KG-IPS estimator satisfies the following confidence interval with probability at least $1 - \delta$:*

$$\left|\hat{V}_{KG\text{-}IPS}(\theta) - V(\theta)\right| \leq O\left(M\sqrt{\log(1/\delta)/n}\right).$$

A detailed analysis of the variance and confidence interval can be found in Appendix B.

To further support our findings, we demonstrate that the optimal policy for the final reward is consistent with the optimal policy for the entity-based knowledge graph reward, which means the non-entity-based LLM reward can be considered as a regularization term that does not affect the optimal

policy. See Appendix C for a complete analysis. In the end, we could directly optimize the target policy by maximizing the estimated value function through policy gradient,

$$\theta \leftarrow \theta + \nabla_\theta \hat{V}_{KG-IPS}(\theta). \tag{4}$$

## 4.2 KNOWLEDGE GRAPH PREFERENCE MODELING

To facilitate the evaluation and optimization, we model knowledge graph preference and derive feedback by developing the behavior policy $\mu_\phi$ which verbalizes knowledge-graph trajectories. Randomly sampled trajectories $\mathbb{H}$ from $\mathcal{G}$ in Section 3.2 contain samples that may not be transformed into a chain of thoughts leading to a reasonable question-answering. Following conventional self-consistent measurement (Wang et al., 2022b; Manakul et al., 2023), given a sampled trajectory $h$ and its verbalized chain of thoughts $c$, we prompt the GPT-4 model to propose a question $q$ related to the first entity $e_0 \in h$ whose answer should be exactly the last entity $e_T \in h$, and query the GPT-4 model with the proposed question,

$$\hat{q} \sim f(q|e_0, e_T, c), \quad \hat{y} \sim f(y|\hat{q}, c), \quad R(h|c) = \mathrm{E}\left[\mathbf{1}\left\{e_T = \hat{y}\right\}\right],$$

where the reward of the trajectory is determined by the answer accuracy. We estimate the reward function $R(h|c)$ as the normalized question-answering accuracy (detailed in Appendix D). Then we fine-tune the preference policy $\mu_\phi$ directly via policy gradient optimization,

$$\nabla_\phi J(\phi) = \nabla_\phi \sum_{k=1}^{K} \sum_{t=0}^{|c_k|-1} R(h_k|c_k) \log \mu_\phi(y_{k,t}|q_k, y_{k,<t}),$$

where $J(\phi)$ denotes the overall objective function representing the expected cumulative reward of the policy. Based on the distribution of relations (Figure 4b) and entities (Figure 4c) in the sampled knowledge graph trajectories, we observe that the relation distribution is relatively more skewed toward the most frequent relations. This suggests that the verbalized knowledge graph reasoning policy is likely to focus on more frequent reasoning transitions, potentially enhancing its ability to learn meaningful patterns. In contrast, the entity distribution shows a relatively short tail, which may help mitigate the risk of overfitting to specific entities or knowledge biases.

## 5 EXPERIMENTS

In this section, we evaluate our proposed method, OCEAN, by conducting chain-of-thought alignment on four LLM backbone models and evaluating several downstream tasks. We show our method's effectiveness in chain-of-thought alignment and its generalizability in various tasks to understand (i) whether the proposed optimization approach sufficiently aligns LLMs' chain-of-thought behaviors with higher estimated values on multi-hop question-answering tasks, (ii) how the proposed method performs on knowledge-intensive question-answering tasks and (iii) whether the post-alignment LLM generalizes on commonsense reasoning tasks.

### 5.1 IMPLEMENTATION DETAILS

**Datasets.** Following Zhang et al., we evaluate our approach on three aspects of question answering. For *knowledge-intensive reasoning*, we use datasets that require deep domain understanding. **ARC** (Clark et al., 2018) tests advanced reasoning with grade-school science questions, **PubMedQA** (Jin et al., 2019) assesses biomedical reasoning from abstracts, and **SciQA** (Auer et al., 2023) challenges models using the Open Research Knowledge Graph. For *multi-hop reasoning*, where models combine multiple sources, we use **HotpotQA** (Yang et al., 2018) (reasoning across Wikipedia articles), **MuSiQue** (Trivedi et al., 2022) (requiring 2-4 inference hops), and **StrategyQA** (Geva et al., 2021) (testing implicit reasoning). For *commonsense reasoning*, we evaluate using three commonsenseQA benchmarks (**CSQA** (Talmor et al., 2021), **CSQA2** (Saha et al., 2018), and CSQA-COT1000 (Li et al., 2024a)), along with **OpenBookQA** (Mihaylov et al., 2018) and **WinoGrande** (Sakaguchi et al., 2021). These tasks test models' general commonsense question-answering abilities.

**Baselines.** We experiment with four backbone LLMs: Gemma-2 (Team, 2024) with 2B model parameters, Llama-3 (AI@Meta, 2024) with 8B model parameters, Phi-3.5-mini (Abdin et al., 2024)

with 3.8B model parameters, and Mistral-0.2 (Jiang et al., 2023) with 7B model parameters. We use the instruction fine-tuned version of backbone LLMs for better instruction following abilities in question-answering. For chain-of-thought alignment in OCEAN, we use the CWQ question-answering dataset (Talmor & Berant, 2018) as the source data, in which the question-answering pairs are developed from knowledge graphs. OCEAN only uses CWQ questions for the LLM to generate chain-of-thought reasoning paths, which are further aligned using the knowledge graph preference model, without directly supervised learning on the ground-truth answers. To compare with direct supervised learning, we also enable instruction-tuning as a baseline (SFT), which is fine-tuned with the question as instruction and the answer as the response.

## 5.2 MULTI-HOP QUESTION ANSWERING

We evaluate the chain-of-thought reasoning performance of OCEAN compared with base LLMs and supervised fine-tuning (SFT), in three multi-hop question-answering tasks in Table 1. Comparing SFT and Base LLMs, we observe similar knowledge inconsistency as in knowledge-intensive tasks. Although SFT improves on MuSiQue with Gemma-2 and Mistral-0.2 backbones whose base models' performance is relatively inferior on this task, such knowledge-inconsistent problems result in worse performance on other downstream tasks.

| Model | Method | HotpotQA | | | MuSiQue | | StrategyQA | | |
|---|---|---|---|---|---|---|---|---|---|
| | | w/ ctx (%) | w/o ctx (%) | $\hat{V}(\theta)$ | w/ ctx (%) | $\hat{V}(\theta)$ | w/ ctx (%) | w/o ctx (%) | $\hat{V}(\theta)$ |
| Llama-3 | Base | 32.78 | 33.54 | -10.35 | 11.59 | -9.90 | 77.73 | 59.53 | -9.25 |
| | SFT | 8.22 (-24.56) | 16.49 (-17.05) | -22.28 | 1.80 (-9.79) | -17.09 | 66.52 (-11.21) | 51.82 (-7.71) | -15.17 |
| | OCEAN | 33.38 (+0.6) | 33.75 (+0.21) | **-8.10** | 11.67 (+0.08) | **-9.77** | 75.40 (-2.33) | 59.83 (+0.3) | **-5.53** |
| Gemma-2 | Base | 26.33 | 18.58 | -31.88 | 5.84 | -26.41 | 76.71 | 60.99 | -14.06 |
| | SFT | 29.75 (+3.42) | 15.91 (-2.67) | -46.92 | **12.53** (+6.69) | -40.25 | 64.77 (-11.94) | 51.97 (-9.02) | -23.27 |
| | OCEAN | 26.20 (-0.13) | 19.70 (+1.12) | **-26.43** | 6.87 (+1.03) | **-22.15** | 74.24 (-2.47) | **66.23** (+5.24) | **-13.52** |
| Phi-3.5 | Base | 32.13 | 26.14 | -19.49 | 11.85 | -15.30 | 73.51 | 58.37 | -13.87 |
| | SFT | 21.99 (-10.14) | 7.87 (-18.27) | -44.57 | 6.01 (-5.84) | -42.10 | 63.03 (-10.48) | 50.95 (-7.42) | -21.66 |
| | OCEAN | **35.13** (+3.0) | 26.23 (+0.09) | **-14.84** | 10.82 (-1.03) | **-13.47** | 72.20 (-1.31) | 57.64 (-0.73) | **-12.25** |
| Mistral-0.2 | Base | 26.82 | 28.13 | -19.08 | 5.67 | -6.40 | **79.33** | 58.22 | -11.36 |
| | SFT | 20.88 (-5.94) | 14.49 (-13.64) | -18.53 | 7.73 (+2.06) | -12.24 | 52.40 (-26.93) | 51.53 (-6.69) | -15.39 |
| | OCEAN | 27.24 (+0.42) | 27.54 (-0.59) | **-3.12** | 5.15 (-0.52) | **-5.94** | 77.29 (-2.04) | 56.62 (-1.6) | **-11.21** |

Table 1: Comparison results of OCEAN, base LLMs (Base), and supervised fine-tuning (SFT), on three **Multi-hop Question-answering** tasks. We report with context (**w/ ctx**) and without context (**w/o ctx**) answer results with the Exact Match (EM) metric on **HotpotQA** and the Accuracy metric on **StrategyQA**. Performance on **MuSiQue** dataset is EM with context. We also use each test/validation split for each dataset and report policy evaluation $\hat{V}(\theta)$ results. We highlight the best-performed metric in **bold font** and the second-best underline for each task.

Since OCEAN is aligned to incorporate more knowledge-faithful chain-of-thought reasoning patterns learned from knowledge graph reasoning policy without directly editing its internal knowledge, OCEAN maintains its generalizability in adapting to downstream tasks. We observe that OCEAN consistently improves on the policy estimated value $\hat{V}(\theta)$ through direct policy optimization proposed in equation 4, which demonstrates the effectiveness of the developed optimization method. Regarding the question-answering accuracy, OCEAN improves base LLMs, which achieves the best performance on HotpotQA and StrategyQA without context.

## 5.3 KNOWLEDGE-INTENSIVE QUESTION ANSWERING

To understand the effectiveness of OCEAN in knowledge-intensive question-answering tasks, we show performance comparison with base LLMs (Base) and supervised fine-tuning (SFT) in Table 2. Comparing SFT and Base LLMs, we observe that directly aligning knowledge graphs with LLMs may suffer from domain and knowledge inconsistency when downstream tasks require specific domain knowledge, conflicting with the knowledge graph in the fine-tuning stage. We also observe that SFT achieves 4.85% and 0.55% average improvements on the PubMedQA dataset, with and without context respectively, whereas it suffers from 29.60%, 8.35%, 13.6% average performance decreases on the remaining tasks. Such significant discrepancies in SFT's effects across different downstream tasks further show the risk in direct knowledge editing in LLMs.

| Model | Method | ARC | | PubMedQA | | | SciQA | | |
|---|---|---|---|---|---|---|---|---|---|
| | | w/o ctx (%) | $\hat{V}(\theta)$ | w/ ctx (%) | w/o ctx (%) | $\hat{V}(\theta)$ | w/ ctx (%) | w/o ctx (%) | $\hat{V}(\theta)$ |
| Llama-3 | Base | 79.93 | **-10.38** | 63.60 | 58.60 | -25.40 | 83.10 | 57.10 | -22.91 |
| | SFT | 61.87 (-18.06) | -18.42 | **75.80** (+12.2) | 58.00 (-0.6) | -26.03 | 67.10 (-16.0) | 35.80 (-21.3) | -23.84 |
| | OCEAN | 80.60 (+0.67) | -12.45 | 66.00 (+2.4) | **59.80** (+1.2) | **-9.37** | 83.20 (+0.1) | 57.70 (+0.6) | **-16.63** |
| Gemma-2 | Base | 65.89 | **-15.36** | 34.40 | 40.60 | -24.61 | 76.50 | 47.10 | **-26.60** |
| | SFT | 18.06 (-47.83) | -25.22 | 35.60 (+1.2) | 21.00 (-19.6) | -26.55 | **79.80** (+3.3) | **51.50** (+4.4) | -36.61 |
| | OCEAN | 63.21 (-2.68) | -16.20 | **44.60** (+10.2) | **41.60** (+1.0) | **-18.72** | 72.20 (-4.3) | 47.50 (+0.4) | -26.77 |
| Phi-3.5 | Base | 87.29 | **-7.86** | 70.40 | 41.80 | -28.48 | 83.50 | 58.90 | -14.46 |
| | SFT | 65.22 (-22.07) | -9.02 | 62.40 (-8.0) | 50.20 (+8.4) | -28.40 | 76.90 (-6.6) | 43.80 (-15.1) | -14.62 |
| | OCEAN | **87.63** (+0.34) | -7.94 | 68.40 (-2.0) | 47.60 (+5.8) | **-11.45** | **84.70** (+1.2) | **63.50** (+4.6) | **-13.40** |
| Mistral-0.2 | Base | 73.91 | **-9.99** | 51.60 | 36.20 | -13.01 | 78.50 | 58.00 | **-11.77** |
| | SFT | 43.48 (-30.43) | -13.99 | **65.60** (+14.0) | **50.20** (+14.0) | -21.87 | 64.40 (-14.1) | 35.50 (-22.5) | -21.86 |
| | OCEAN | 68.90 (-5.01) | -10.89 | 52.60 (+1.0) | 33.20 (-3.0) | **-12.42** | **79.10** (+0.6) | 58.40 (+0.4) | -12.00 |

Table 2: Comparison results of `OCEAN`, base LLMs (`Base`), and supervised fine-tuning (`SFT`), on three **Knowledge-intensive Question-answering** tasks. We report answers with context (**w/ ctx**) and without context (**w/o ctx**) on Exact Match (EM) metric on **PubMedQA** and **SciQA**. The EM performance on **ARC** dataset is without context. We also use the test/validation split for each dataset to report estimated policy values $\hat{V}(\theta)$. We highlight the best metric in **bold font** for each task.

With the enhancement of `OCEAN`, question-answering accuracy of knowledge-intensive tasks generally improved, while `OCEAN` fine-tuned LLMs achieving the best performance on all three datasets, except for PubMedQA without context where SFT achieves better performance due to knowledge transfer from knowledge graph dataset. We also observe consistent policy value improvement on PubMedQA and SciQA, where the original policy values of base LLMs are relatively lower. For tasks like ARC, which does not require additional reference knowledge from context and reasoning in an easier chain of thought, `OCEAN` still maintains comparable policy value to the base LLM, which demonstrates the robustness and generalizability of the proposed method.

## 5.4 COMMONSENSE REASONING

| Model | Method | CSQA | CSQA-2 | CSQA-COT1000 | OpenBookQA | Winogrande | Average |
|---|---|---|---|---|---|---|---|
| Llama-3 | Base | **65.03** | **71.39** | 69.50 | 58.80 | **43.09** | **61.56** |
| | SFT | 51.19 (-13.84) | 57.06 (-14.33) | 49.00 (-20.5) | 63.20 (+4.4) | 34.73 (-8.36) | 51.04 |
| | OCEAN | **65.03** (0.0) | 68.60 (-2.79) | **72.00** (+2.5) | **60.40** (+1.6) | 41.36 (-1.73) | 61.48 |
| Gemma-2 | Base | 57.99 | 62.57 | 63.50 | 51.80 | 49.64 | 57.10 |
| | SFT | 14.66 (-43.33) | 65.80 (+3.23) | 15.00 (-48.5) | 7.40 (-44.4) | 50.04 (+0.4) | 30.58 |
| | OCEAN | **67.73** (+9.74) | **63.56** (+0.99) | **72.50** (+9.0) | **57.20** (+5.4) | **50.12** (+0.48) | **62.22** |
| Phi-3.5 | Base | 68.55 | **64.70** | 72.50 | **72.40** | **50.51** | **65.73** |
| | SFT | 69.94 (+1.39) | 61.47 (-3.23) | 72.00 (-0.5) | 69.40 (-3.0) | 50.51 (0.0) | 64.66 |
| | OCEAN | **69.62** (+1.07) | 62.77 (-1.93) | **73.50** (+1.0) | 71.20 (-1.2) | 50.12 (-0.39) | 65.44 |
| Mistral-0.2 | Base | 61.18 | 68.48 | 65.00 | **64.00** | 46.25 | 60.98 |
| | SFT | 35.87 (-25.31) | 22.47 (-46.01) | 33.00 (-32.0) | 34.80 (-29.2) | 29.12 (-17.13) | 31.05 |
| | OCEAN | **63.80** (+2.62) | **69.19** (+0.71) | **67.00** (+2.0) | 62.60 (-1.4) | **46.49** (+0.24) | **61.82** |

Table 3: Comparison results of `OCEAN`, base LLMs (`Base`), and supervised fine-tuning (`SFT`), on five **Commonsense Reasoning** tasks. We report the Exact Match (EM) metric on these tasks and the average performance. We highlight the best method in **bold font** for each task and LLM.

Finally, to demonstrate `OCEAN`'s generalizability in preserving commonsense knowledge and preventing knowledge catastrophic forgetting (Luo et al., 2023; Wu et al., 2025), we evaluate `OCEAN` with base LLMs (`Base`) and supervised fine-tuning (`SFT`) on five commonsense reasoning tasks in Table 3. Since such tasks do not require external domain knowledge, we only evaluate the accuracy of the model's generated answers. We observe that directly applying supervised fine-tuning (SFT) using knowledge graphs significantly impacts large language models (LLMs), potentially leading to catastrophic forgetting of commonsense knowledge. especially for the backbone LLMs of Gemma-2 and Mistral-0.2. In contrast, we show that `OCEAN` achieves robust performance on commonsense reasoning by leveraging off-policy evaluation and optimization from knowledge graph's feedback. `OCEAN` manages to maintain comparable performance of base LLMs (*e.g.*, Llama-3 and Phi-3.5),

which have strong zero-shot commmonsense reasoning abilities. In addition, we observe that for base LLM with relatively lower performance (*e.g.*, Gemma-2 and Mistral-0.2), OCEAN enables consistent improvements. Therefore, OCEAN serves as a robust off-policy alignment paradigm to incorporating knowledge graph reasoning without affecting the generalizability of pretrained LLMs.

## 6 ANALYSIS

### 6.1 IN-CONTEXT LEARNING & INSTRUCTION TUNING

We conduct further analysis to compare the performance of both the base model and our proposed model in the scenarios of In-Context Learning and instruction fine-tuning. Specifically, we conduct this analysis using the Gemma-2 and Phi-3.5 models across three benchmark datasets: SST2 (Socher et al., 2013) for sentiment classification, AgNews (Zhang et al., 2015) for topic classification, and BoolQ (Clark et al., 2019) for reading comprehension. In the In-context Learning setup, we provide the model with a single example for each task in the prompt. For the Instruction tuning experiments, we apply LoRA (Hu et al., 2021) to the pre-trained model and fine-tune it on each dataset for 10 epochs. Throughout these experiments, the rank parameter in LoRA is fixed at 16, and we set $\alpha$ in LoRA to 32 across all tasks. The results of the In-context Learning and Instruction Tuning are presented in Table 4. Overall, we observe that the performance of the base model and our proposed model is largely comparable across most scenarios, except in the AG News task with Gemma-2, where OCEAN demonstrates greater performance after instruction tuning.

| Model | Method | In-Context Learning | | | | Instruction-Tuning | | | |
|---|---|---|---|---|---|---|---|---|---|
| | | SST2 | BoolQ | AG News | Avg. | SST2 | BoolQ | AG News | Avg. |
| Gemma-2 | Base | 87.16 | 56.12 | 16.47 | 53.25 | 96.21 | 69.03 | 47.03 | 70.76 |
| | OCEAN | 89.33 | 55.72 | 13.14 | 52.73 | 96.56 | 68.66 | 60.08 | 75.10 |
| Phi-3.5 | Base | 41.28 | 60.06 | 31.89 | 44.41 | 96.44 | 68.13 | 86.43 | 83.67 |
| | OCEAN | 40.48 | 59.11 | 32.37 | 43.98 | 96.44 | 68.81 | 86.24 | 83.83 |

Table 4: Performance Comparison of In-Context Learning and Instruction Tuning. All datasets consist of classification tasks or true/false questions, so accuracy is used to evaluate the performance. The performance of the base model and our proposed model is largely comparable.

### 6.2 EVALUATION OF GENERATION QUALITY POST ALIGNMENT

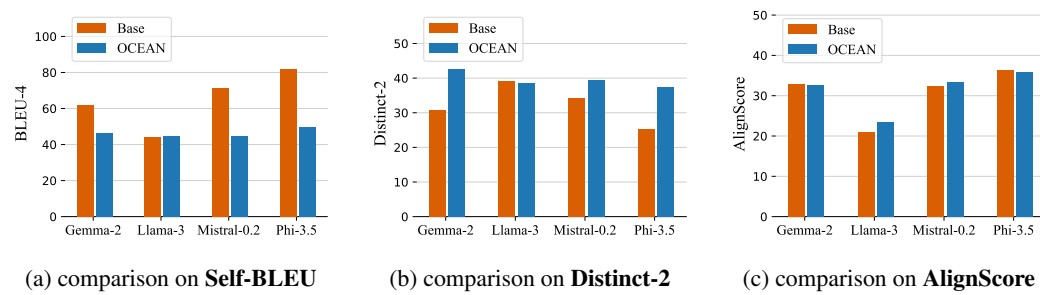

(a) comparison on **Self-BLEU**   (b) comparison on **Distinct-2**   (c) comparison on **AlignScore**

Figure 2: Comparison results of base LLMs and OCEAN on three evaluation metrics, Self-BLEU, Distinct-2, and AlignScore. Lower Self-BLEU scores and higher Distinct-2 scores indicate better diversity of the generated text, while higher AlignScore indicates better faithfulness.

To further evaluate the generation quality of post-alignment LLMs, we use the *Self-BLEU* (Zhu et al., 2018) and *Distinct-2* (Li et al., 2015) scores to evaluate the diversity of the generation, concerning the similarity between generated texts and the uniqueness of generated 2-gram phrases respectively. In addition, *AlignScore* (Zha et al., 2023) is used to evaluate the faithfulness of the generated answer given the question context. The results are presented in Figure 2, which show that post-alignment LLMs achieve comparable or better performances in terms of generation diversity and faithfulness.

This demonstrates that while OCEAN aligns chain-of-thought reasoning with KGs, we maintain the text generation qualities of LLMs.

## 6.3 CASE STUDY

In the previous Section 5.2 and 5.3 we observe efficient chain-of-thought alignment with improvement on the estimated policy value $\hat{V}(\theta)$. To further understand the effects of the alignment, we choose backbone LLMs, Gemma-2 and Llama-3, with significant improvements on $\hat{V}(\theta)$, and perform a sample analysis by comparing the outputs of the base model and OCEAN on the same set of questions. Our findings demonstrate that the application of our method enhances the precision and conciseness of the chain of thought in the generated responses. Some illustrative examples are provided in Figure 3. Specifically, in the first example, the base Llama-3 model incorrectly claims that singing is not a primary action associated with playing the guitar, which leads to an erroneous solution to the question. In contrast, our method enables the model to recognize that singing is a common activity when playing the guitar, while also understanding that making music serves as a broader term. In the second example, although both the base model and OCEAN on Gemma-2 provide reasonable answers to the question, our model demonstrates a more concise chain of thought, streamlining the reasoning process and arriving at the solution with greater simplicity.

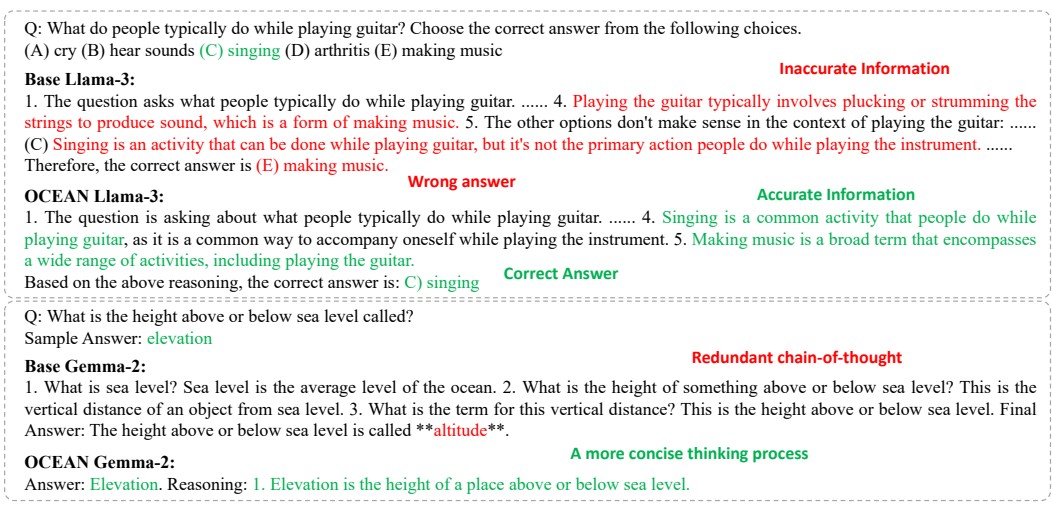

Figure 3: Sample comparison between the base model and OCEAN on Llama-3 and Gemma-2. Our method enables a more precise and concise chain of thought.

## 7 CONCLUSION

In this work, we propose OCEAN to address the challenge of offline chain-of-thought evaluation and optimization of LLMs. By modeling the knowledge-graph preference and deriving feedback by developing a policy that verbalizes knowledge-graph trajectories, we propose KG-IPS estimator to estimate policy values in the alignment of reasoning paths with knowledge graphs. Theoretically, we proved the unbiasedness of the KG-IPS estimator and provided a lower bound on its variance. Empirically, our framework effectively optimizes chain-of-thought reasoning while maintaining LLMs' general downstream task performance, offering a promising solution for enhancing reasoning capabilities in large language models. Our framework not only enhances chain-of-thought reasoning but can also offer a potential offline evaluation mechanism for agentic frameworks, enabling the safe assessment of autonomous decision-making processes. Future work could explore integrating this approach into reinforcement learning and multi-agent systems to further validate its utility in complex, dynamic environments.

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

# A   MEAN ANALYSIS

To prove that the KG-IPS estimator is unbiased, we need to demonstrate that the expected value of the IPS estimator equals the true expected reward under $\pi_\theta$.

*Proof.* The value function of policy $\pi_\theta$ can be defined as:

$$V(\pi_\theta) = \mathbb{E}_{a_t \sim \pi(\cdot|s_t)} \left[ r(s_t, a_t) \right]$$
$$= \frac{1}{N} \sum_{i=1}^{N} \frac{1}{T_i |c_t^{(i)}|} \sum_{t=1}^{T_i} \mathbb{E}_{e \sim \pi_\theta(\cdot|s_t^{(i)})} \left[ r(s_t^{(i)}, e) \right]$$

where $r(s_t, a_t)$ represent the reward obtained by taking action $a_t$ under state $s_t$.

Given that our value function consists of two cases: the first case considers the reward derived from the entity tokens under the knowledge graph preference policy $\mu_\theta$, and the second case focuses on the reward derived from the non-entity tokens under the base LLM policy $\pi_0$. We separately prove the unbiasedness by showing that the expected value of either the entity-based or non-entity-based estimators is equal to the true expected reward under their respective policies.

The expected value of the entity tokens in the knowledge graph is:

$$\mathbb{E}\left[\hat{V}_{KG}(\theta)\right] = \frac{1}{N}\sum_{i=1}^{N}\frac{1}{T_i|c_t^{(i)}|}\sum_{t=1}^{T_i}\mathbb{E}_{e\sim\mu_\phi(\cdot|s_t^{(i)}),e\sim\mathcal{P}(e)}\left[\frac{\pi_\theta(e|s_t^{(i)})}{\mu_\phi(e|s_t^{(i)})}\log\pi_0(e|s_t^{(i)})\right]$$

$$= \frac{1}{N}\sum_{i=1}^{N}\frac{1}{T_i|c_t^{(i)}|}\sum_{t=1}^{T_i}\mathbb{E}_{e\sim\mu_\phi(\cdot|s_t^{(i)})}\left[\frac{\pi_\theta(e|s_t^{(i)})}{\mu_\phi(e|s_t^{(i)})}\mathbb{P}(e=\hat{y}|s_t^{(i)},e)\right]$$

$$= \frac{1}{N}\sum_{i=1}^{N}\frac{1}{T_i|c_t^{(i)}|}\sum_{t=1}^{T_i}\mathbb{E}_{e\sim\pi_\theta(\cdot|s_t^{(i)})}\left[\mathbb{P}(e=\hat{y}|s_t^{(i)},e)\right]$$

$$= \frac{1}{N}\sum_{i=1}^{N}\frac{1}{T_i|c_t^{(i)}|}\sum_{t=1}^{T_i}\mathbb{E}_{e\sim\pi_\theta(\cdot|s_t^{(i)})}\left[r(s_t^{(i)},e)\right] = V(\pi_\theta)$$

For non-entity tokens, the proof is similar:

$$\mathbb{E}\left[\hat{V}_{base}(\theta)\right] = \frac{1}{N}\sum_{i=1}^{N}\frac{1}{T_i|c_t^{(i)}|}\sum_{t=1}^{T_i}\mathbb{E}_{e\sim\pi_0(\cdot|s_t^{(i)}),e\sim\mathcal{P}(e)}\left[\frac{\pi_\theta(e|s_t^{(i)})}{\pi_0(e|s_t^{(i)})}\log\pi_0(e|s_t^{(i)})\right]$$

$$= \frac{1}{N}\sum_{i=1}^{N}\frac{1}{T_i|c_t^{(i)}|}\sum_{t=1}^{T_i}\mathbb{E}_{e\sim\pi_0(\cdot|s_t^{(i)})}\left[\frac{\pi_\theta(e|s_t^{(i)})}{\pi_0(e|s_t^{(i)})}\mathbb{P}(e=\hat{y}|s_t^{(i)},e)\right]$$

$$= \frac{1}{N}\sum_{i=1}^{N}\frac{1}{T_i|c_t^{(i)}|}\sum_{t=1}^{T_i}\mathbb{E}_{e\sim\pi_\theta(\cdot|s_t^{(i)})}\left[\mathbb{P}(e=\hat{y}|s_t^{(i)},e)\right] = V(\pi_\theta)$$

This completes the proof. □

## B VARIANCE ANALYSIS

In this section, we derive a confidence bound from the confidence interval and calculate the lower bound of the variance for the KG-IPS estimator.

*Proof.* Let $M$ be the maximum value of $\frac{\pi_\theta(e|s_t)}{\mu_\phi(e|s_t)}$ ranging over all entity tokens $e$. This quantity $M$ represents the largest discrepancy between the target policy $\pi_\theta$ and the behavior policy $\mu_\phi$.

Since each reward $r_t$ lies in $[0,1]$, it is $\frac{1}{4}$-sub-Gaussian. Consequently, multiplying $r_t$ by a constant factor of at most $M$ produces a random variable of the form

$$X_t = \frac{\pi_\theta(e|s_t)}{\mu_\phi(e|s_t)}r_t,$$

that is $(\frac{M^2}{4})$-sub-Gaussian. Let $\hat{V}_{\text{KG-IPS}}(\theta)$ be the average of $n$ i.i.d. samples $\{X_t\}_{t=1}^{n}$. From the property of sub-Gaussian variables, if each $X_t$ is $(\frac{M^2}{4})$-sub-Gaussian, then the average

$$\hat{V}_{\text{KG-IPS}}(\theta) = \frac{1}{n}\sum_{t=1}^{n}X_t$$

is $(\frac{M^2}{4n})$-sub-Gaussian. In particular, this implies that the variance cannot be smaller than $\Omega(\frac{M^2}{n})$, indicating an irreducible noise level of order $\frac{M}{\sqrt{n}}$.

For any sub-Gaussian random variable $X$ with variance $\sigma^2$, the concentration inequality (Zhang & Chen, 2020) holds:

$$\left|\hat{X} - \mathbb{E}[X]\right| \leq \sigma\sqrt{2\log\left(\frac{1}{\delta}\right)}.$$

Plugging the variables above into the inequality, we get the following bound for the KG-IPS estimator:

$$\left|\hat{V}_{\text{KG-IPS}}(\theta) - V(\theta)\right| \leq M\sqrt{\frac{\log(1/\delta)}{2n}} = O(M\sqrt{\log(1/\delta)/n}),$$

with probability at least $1 - \delta$.

The variance of a sub-Gaussian random variable is close to its variance proxy, which means the lower bound on the variance of the KG-IPS estimator is $\Omega(\frac{M^2}{n})$. In addition, by standard concentration inequalities, we can get $O(M\sqrt{\log(1/\delta)/n})$ confidence intervals on our estimator for policy $\pi_\theta$. □

## C  THEORETICAL ANALYSIS

The value function of policy $\pi_\theta$ is defined as:

$$V^{\pi_\theta}(s_t, a_t) = \mathbb{E}_{a_t \sim \pi(\cdot|s_t)}\left[r(s_t, a_t)\right].$$

Based on our settings, we optimize the target policy for entity tokens aligning with knowledge graph preference. The reward function is formulated as:

$$r^{\text{KG}}(s_t, a_t) = \sum_{e \in a_t} \frac{\pi_\theta(e|s_t)}{\mu_\phi(e|s_t)} \log \pi_0(e|s_t), \tag{5}$$

where $\mu_\phi$ is the knowledge graph preference policy.

To reduce variance, the logged rewards for non-entity tokens under the base LLM policy $\pi_0$ are incorporated as a regularization term in the reward function, formulated as:

$$r^{\text{reg}}(s_t, a_t) = \sum_{v \in c_t \setminus a_t} \frac{\pi_\theta(v|s_t)}{\pi_0(v|s_t)} \log \pi_0(v|s_t), \tag{6}$$

where $\pi_0$ is the base LLM policy. This helps mitigate disturbances, ensuring the LLM's behavior on non-entity tokens remains stable and preventing model degeneration.

The final reward is:

$$r(s_t, a_t) = r^{\text{KG}}(s_t, a_t) + r^{\text{reg}}(s_t, a_t), \tag{7}$$

Since both $r^{\text{KG}}(s_t, a_t)$ and $r^{\text{reg}}(s_t, a_t)$ are reweightings of the log-based reward $\log \pi_0(v|s_t)$, they belong to the same equivalence class. By leveraging Lemma 2 from DPO (Rafailov et al., 2024), we show that the optimal policy for the task-specific reward $r^\phi$ aligns with the optimal policy for the final reward $r$. This implies that both rewards induce the same optimal policy.

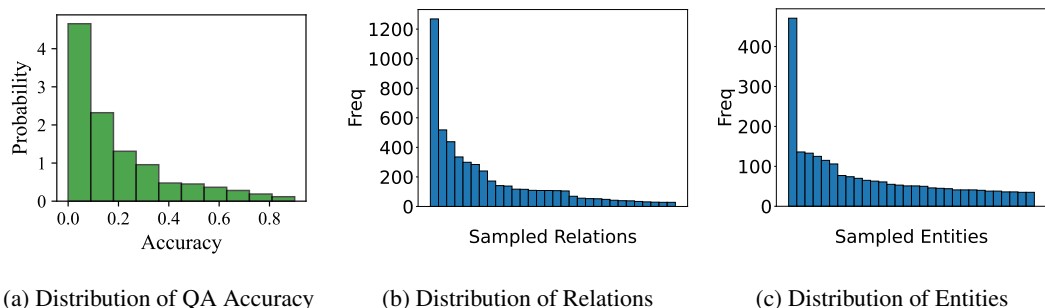

(a) Distribution of QA Accuracy     (b) Distribution of Relations     (c) Distribution of Entities

Figure 4: Sampling distributions of (a) trajectories in the knowledge graph that are verbalized as multi-step QA tasks and successfully answered by the LLM itself, (b) relations, and (c) entities in the knowledge graphs and their frequencies of the appearance in the trajectories sampled from the *Wikidata5M* (Wang et al., 2021) knowledge graph.

## D  DETAILS OF KNOWLEDGE GRAPH PREFERENCE MODELING

The verbalized trajectories have in average 141.64 tokens with a standard deviation of 34.39. The average trajectory length is 4.79 steps with a standard deviation of 0.56. For each trajectory, there are in average 5.79 entities with a standard deviation of 0.56, and 4.11 relations with a standard

deviation of $0.90$. In Figure 4a, we present the probability distribution of sampled trajectories, with respect to the number of correct answers generated per trajectory from ten differently sampled questions associated with each trajectory. Based on such self-consistency measurement, we estimate the reward function $R(\boldsymbol{h}|\boldsymbol{c})$ as the normalized question-answering accuracy.

**Knowledge Graph Preference Model.** The knowledge graph preference model is developed based on the pre-trained GPT2-Medium model (Radford et al., 2019). We collected 6K question-answering pairs from the Wikidata5M (Wang et al., 2021) knowledge graph based on the sampling strategy in Section 4.2. The sampled knowledge graph trajectories are composed into natural language prefixed by the corresponding questions by the GPT-4 model, which verbalizes the knowledge graph reasoning trajectories and aligns with generative language models' behaviors. The model is then fine-tuned with a base learning rate of $1e-4$ for 10 epochs with a linear learning scheduler.

