# OpenReview forum: "OCEAN: Offline Chain-of-thought Evaluation and Alignment in Large Language Models"
_ICLR.cc/2025/Conference — ICLR 2025 Poster_

### Official Review · Reviewer_3w4e · 2024-10-31

**Soundness:** 3
**Presentation:** 3
**Contribution:** 3
**Rating:** 6
**Confidence:** 4

**Summary:**

This paper introduces a novel approach for aligning LLMs with chain-of-thought reasoning by leveraging offline knowledge graph data. Specifically, a knowledge graph preference policy is first trained using reinforcement learning on question-answering data derived from verbalized exploration trajectories within the offline knowledge graph. This policy, which serves as a proxy for the knowledge graph’s preferences, is then used to fine-tune the target LLM policy through a novel KG-IPS estimator that assesses the difference between the proxy knowledge graph preference policy and the target policy. Extensive experiments are conducted to evaluate both the chain-of-thought reasoning alignment performance and the impact of this alignment on the base LLM's general capabilities.

**Strengths:**

- The research goal of aligning LLMs with chain-of-thought reasoning is both interesting and important.
- The approach of using graphs as supplementary information to align LLMs with chain-of-thought reasoning is a promising avenue for exploration.
- The paper is clearly presented and easy to follow, making it a pleasure to read overall.
- The experiments are thorough and extensive.

**Weaknesses:**

Several key concepts and notations in the task formulation and methodology are not clearly defined, and certain details about the dataset are missing. Please refer to the questions for further details.

**Questions:**

# Task formulation
- Line 156, what is the "surrounding context $c_t  \backslash a_t$"? Could you please elaborate on why $a_t$ "is a sub-sequence in the reasoning path $a_t \in c_t$,"? Providing some intuitive examples may help improve understanding.
- In line 169, "the current edge $e_{t-1}$" implies that $e$ represents edges in graphs. However, in line 172, "the entity $e_{t-1}$" implies that it represents entities. Could you please clarify the meaning of the notation $e$? The notation $e$ also appears in Equation 3. Please see related questions below.

# Methodology
- As most of readers would be not familiar with the Wikidata5M dataset, could you please present some cases of sampled trajectories $h$ from the knowledge graph $\mathcal{G}$? In addition, some cases of the verbalizing transformation from $h$ to $c$ using GPT-4 would also be very interesting to see.
- What are the transition dynamics in the exploration of the knowledge graph?
- Could you please provide more detailed information about the resulting offline dataset used for knowledge graph preference modeling?
- Could you please clarify whether the training of the knowledge graph preference policy is conducted in an online or offline manner?
- About Equation 3:
  - $e \in c_t$ implies that $e$ is a sub-set of $c_t$. Could you please explain a little bit on the relation between these three notations: $e$, $a_t$, and $c_t$? Could we simply understand that $e$ represents a sequence of tokens?
  - $c_t^{(i)}$ is outside of the summation $\sum\limits_{t=1}^{T_i}$. The subscript $t$ for this variable is unassigned.
  - Should $e \in c_t$ be $e \in c_t^{(i)}$?
- Line 221, is it correct that the reward function is defined as $r_t^{(i)} = \log \pi_0(e|s_t^{(i)})$ for the training of the target policy? If so, why we consider the reward in the token level, instead of the reward in the reasoning step level? Utilizing the reward function at the token level conflicts with the formulation in line 159, where $r_t = r(s_t, c_t)$.

# Minor suggestions
- Some background knowledge about the conventional inverse propensity scores (IPS) method could be interesting for readers.

---

> ### Author Response · Authors · 2024-11-22
> **Rebuttal to Reviewer 3w4e (Part 1/2)**
>
> Thank you for your valuable feedback and the time you have spent reviewing our work. We address the concerns raised and provide answers to your questions accordingly.
>
> **Response to Task Formulation 1**
>
> Thank you for your valuable feedback. We will address the questions and ensure clearer definitions of key concepts, notations, and dataset details in the revised version.
>
> Surrounding context refers to textual outputs that do not contribute directly to reasoning but are generated to ensure natural language coherence. For example, in the sample shown in Figure 3 (Above - OCEAN Llama-3), the reasoning path is “playing guitar -$>$ singing.” When combined with the surrounding context, the sentence becomes: “Singing is a common activity that people do while playing guitar.” Here, phrases like “is a common activity that people do while” are not directly related to the reasoning process but are included to maintain fluency and coherence in natural language.
>
> **Response to Task Formulation 2**
>
> We acknowledge the inconsistency in our notation.
> To clarify, in line 169, $e_{t-1}$ should represent the current entity, not the edge. Throughout the rest of the paper, including Equation 3 and line 172,  $e$ consistently represents an entity. We will revise line 169 accordingly.
>
>
> **Response to Methodology 1**
>
> Thank you for the suggestion! Here is an example from our method for your reference. We will also incorporate additional illustrative examples in the paper. In Wikidata5M, one such trajectory is: [“United States”, “contains”, “Arizona (state)”], where “United States” and “Arizona (state)” are the nodes, and “contains” represents their relationship. Following the verbalizing transformation, we generated the paragraph below, where answering the question requires knowledge of this trajectory:
>
> ```
> The North Carolina division of the World Commission on North Carolina Geography (WCNG) is located within the North Carolina tourism department. The state of North Carolina is headed by its governor, Roy A. Cooper III, who is a member of the Democratic Party. North Carolina is a part of the United States of America, which is a democratic country. The state shares its boundaries with the state of Arizona in the United States. Based on the above context, ask a question about WCNG and whose answer should be exactly arizona (u.s. state). The question is: Which state shares its boundaries with North Carolina, as determined by the World Commission on North Carolina Geography (WCNG)? Provide the answer in 5 words:
> ```
>
> **Response to Methodology 2**
>
> We explore the knowledge graph by randomly sampling trajectories, which are then verbalized as described in Section 3.2. These verbalized trajectories are subsequently utilized in the KG-IPS estimator, which offers an unbiased estimate of the target policy and serves as a guide for the LLM fine-tuning process.
>
>
> **Response to Methodology 3**
>
> We provided detailed information about the offline dataset used for preference modeling and the implementation details in Section 5.1 (Line 322 - 327). The offline dataset comprises 6K sampled trajectories from the knowledge graph. We would provide further clarifications or additional details if needed.
>
>
> **Response to Methodology 4**
>
> As explained in Section 4.2, the preference policy is trained with the collected offline dataset using the policy gradient method in Line 273.
> Detailed training and implementation information is provided in Section 5.1 (Line 322 - 327). We would provide further clarifications or additional details if needed.
>
>
> **Response to Methodology 5**
>
> In this equation, $c$ represents the entire reasoning path, which consists of a sequence of tokens. $e$ denotes each individual token within this path.
> Since not all tokens are included in the knowledge graph trajectories, $a_t$ is used to represent the set of entity tokens.
> Thanks for pointing this out. The term $1/ c_t^{(i)}$ should be inside the second summation. Additionally, it should be written as $e \in c_t^{(i)}$. We will update the equation accordingly in our paper.

---

> ### Author Response · Authors · 2024-11-22
> **Rebuttal to Reviewer 3w4e (Part 2/2)**
>
> **Response to Methodology 6**
>
> The reward function in our paper focuses on token-level rewards. Specifically, in the estimation of inverse propensity score (IPS), we need to calculate the ratio $\frac{\pi(e|s_t^{(i)})}{\lambda(e|s_t^{(i)})}$ measuring the entity-level alignment of taking action between the learning and logging policies. Thus, we derive rewards for entity tokens using the knowledge graph preference policy $\mu_\theta$, while rewards for non-entity tokens are based on the base LLM policy $\pi_0$.  Such rewards on tokens instead of steps enable a more accurate estimation of IPS. In addition, fine-grained feedback or supervision maintains the LLM's generation ability and enables more controllable alignment [a,b,c,d]. The formulation in line 159 provides a more general reward function. Our reward function can be viewed as a specific case of the general reward, as it optimizes the policy based on token-level rewards.
>
>
> **Response to Minor suggestions**
>
> Thanks for the suggestions. We will include preliminaries about IPS and offline policy evaluation in our paper.
>
>
> [a] Xu, Dehong, et al. "Aligning Large Language Models via Fine-grained Supervision." arXiv preprint arXiv:2406.02756 (2024).
>
> [b] Yoon, Eunseop, et al. "TLCR: Token-Level Continuous Reward for Fine-grained Reinforcement Learning from Human Feedback." Findings of the Association for Computational Linguistics ACL 2024. 2024.
>
> [c] Li, Wendi, et al. "Reinforcement Learning with Token-level Feedback for Controllable Text Generation." Findings of the Association for Computational Linguistics: NAACL 2024. 2024.
>
> [d] Zeng, Yongcheng, et al. "Token-level Direct Preference Optimization." Forty-first International Conference on Machine Learning.

---

> > ### Comment · Reviewer_3w4e · 2024-11-27
> > **Thanks for your responses**
> >
> > Thank you for the clarification. My concerns have been addressed satisfactorily. Here are some final suggestions to improve the paper:
> >
> > Regarding Methodology 3, I would suggest including more statistical details on the 6K sampled trajectories. Additionally, I recommend that the authors clarify my previous concerns in the revised version, particularly those related to Task Formulation 1 and Methodology 6. This would significantly improve the clarity of both the task formulation and the proposed method.

---

> > > ### Author Response · Authors · 2024-11-27
> > > **Thanks for your responses**
> > >
> > > We sincerely thank you for your thoughtful follow-up feedback, which has been invaluable in improving and clarifying our work.
> > >
> > > **We provide more statistical details on the 6K sampled trajectories.** The verbalized trajectories have in average $141.64$ tokens with a standard deviation of $34.39$. The average trajectory length is $4.79$ steps with a standard deviation of $0.56$. For each trajectory, there are in average $5.79$ entities with a standard deviation of $0.56$, and $4.11$ relations with a standard deviation of $0.90$.
> > >
> > > We will include such statistical details in our paper as suggested. In addition, we will include such clarifications in our revised version to improve the clarity of the task formulation and our method.
> > >
> > > **Please let us know if there are any additional questions, and we are happy to discuss further.**

---

### Official Review · Reviewer_uYKA · 2024-11-04

**Soundness:** 3
**Presentation:** 2
**Contribution:** 2
**Rating:** 6
**Confidence:** 3

**Summary:**

Evaluating the chain-of-thought (CoT) reasoning of large language models (LLMs) is challenging due to the difficulty in gathering accurate labels. Since CoT reasoning involves a sequential decision-making process, using traditional feedback methods like reinforcement learning from human feedback (RLHF) becomes impractical. The author leverages knowledge graphs to address the challenge of evaluating LLMs CoT offline. The authors propose aligning LLMs' multi-step reasoning with multi-hop knowledge graph trajectories using inverse propensity scores (IPS). Specifically, the author proposes OCEAN, an estimator that evaluates a model's reasoning ability using CoTs from knowledge graphs. The authors demonstrated that their proposed offline evaluation method performed well across several reasoning tasks, including multi-hop question answering, knowledge-intensive question answering, and commonsense reasoning.

**Strengths:**

- The authors are focused on addressing a very important problem.
- The authors performed experiments across three types of datasets and four different models.
- The idea of evaluating multi-reasoning with a multi-hop knowlege graph is interesting.

**Weaknesses:**

- Reasoning only requires the model to generate sequences of thoughts with itself, which means we can optimize these thoughts online rather than offline [1,2,3].
- The ability to evaluate a model of multi-step reasoning based only on a multi-hop knowledge graph is very limiting.
- The paper lacks ablation experiments to clearly explain the performance gains of OCEAN, which come from the objective in equation 3 and training on GPT-4's verbalized structured chain of thoughts.

**Questions:**

- How does the proposed IPS approach compare to simply doing the direct method (DM) for evaluation?
- What is the offline policy evaluation performance without performing updated according to equation 4? How does the offline policy evlauation change as you update the policy using equation 4?
- How does SFT + normal COT perform?
- How are you computing the value function estimate for SFT and the base model?
- Did you start training OCEAN with the SFT model?
- Did you also train SFT on the reasoning steps, or just on the questions and answers? If it was only on questions and answers, how does it perform when trained on reasoning steps similar to OCEAN?
- How does the index k depend on the question q and answer y for the policy gradient?
- How does SFT + normal CoT perform?
- What does context mean in w/ctx and w/o ctx?

[1] Rewarding Progress: Scaling Automated Process
Verifiers for LLM Reasoning by Setlur et al.
[2] Quiet-STaR: Language Models Can Teach Themselves to
Think Before Speaking by Zelikman et al.
[3] V-STaR: Training Verifiers for Self-Taught Reasoners by Hosseini et al.

---

> ### Author Response · Authors · 2024-11-22
> **Rebuttal to Reviewer uYKA (Part 1/2)**
>
> Thank you for your valuable feedback and the time you have spent reviewing our work. We address the concerns raised and provide answers to your questions accordingly.
>
>
> **Response to Weakness 1**
>
> As discussed in the Introduction (Line 38 - 47) and Related Work (Line 87 - 103), in general, offline policy evaluation and optimization are important, since online deploying the policy to interact with users can be risky, costly, and impractical [a,b,c], especially for LLMs [k,l].
> Particularly in our setting, online policy optimization requires multi-step feedback on any possible generated sequences of thoughts, which is exhaustive and unrealistic to collect from humans in practice. Besides, human feedback on chain-of-thought reasoning [d] is usually challenging to collect, as detailed in Line 48 - 53 of our paper.
>
> To enable online learning in the knowledge-intensive tasks that are the focus of our paper, we can potentially use the recent work [e] leveraging knowledge graphs to enable evaluating chain-of-thought. However, [e] is restrained to the assumption of accurate chain-of-thought grounding on specific knowledge graphs.
> Specifically, due to the heterogeneity between LLM reasoning and knowledge graph structures, direct interaction and feedback from knowledge graphs on LLM behavior are challenging,
> as they require accurate entity linking and grounding of LLM-generated chains of thought in the knowledge graph.
> To address the above challenge, we develop a verbalized knowledge graph reference model as detailed in Section 4.2.
>
>
> **Response to Weakness 2**
>
> Several recent works have leveraged knowledge graphs to align LLM reasoning [g,h,i,j] with more faithful multi-hop reasoning, which demonstrates the effectiveness and validity of evaluating LLM multi-step reasoning based on knowledge graphs.
> However, different from previous works focusing on retrieval augmentation [g,h] or knowledge injection [i,j], our paper focuses on calibrating LLM reasoning through offline policy evaluation and optimization.
>
>
> **Response to Weakness 3**
>
> As detailed in Line 257 - 280, the usage of GPT-4 is used as an instantiation and limited only to verbalizing structured knowledge graph trajectories into natural language, which is a standard approach in many knowledge graph-enhanced LLM generation methods [m,n,o,p].
>
> During the self-consistent measurement process (Line 257 - 264), the generated question-answering pairs will not be used in LLM fine-tuning, but only for self-consistent sample filtering purposes (such that no extra knowledge is introduced during knowledge graph verbalization). We would like to emphasize that the contributions of our paper are the offline evaluation framework for LLM chain-of-thought generation, direct policy optimization method, and theoretical analysis of the proposed estimator, where the performance improvements come from the offline policy evaluation and optimization method.
>
>
> **Response to Question 1**
>
> Due to the heterogeneity between knowledge graphs and LLM reasoning forms (detailed in Line 60 - 64 of our paper),
> a Direct Method (DM) estimator trained on knowledge graphs cannot directly evaluate complex LLM-generated sequences of thoughts, which leads to limited reward feedback on multi-step reasoning paths and thus Direct Method (DM) also will be biased in estimation [f,q,r,s,t].
>
> Intuitively, such heterogeneity can be addressed by the Equation in line 207 of our paper. the ratio $\frac{\pi(e|s_t^{(i)})}{\lambda(e|s_t^{(i)})}$ measures the entity-level alignment of taking an action between the learning and logging policies.
> Without considering this term, DM can not handle the heterogeneity.
>
>
> **Response to Question 6**
>
> The baseline SFT is just trained on the questions and answers. It is not practical/scalable to train SFT on reasoning steps, for the following considerations. First, it is not scalable to conduct SFT on the reasoning steps collected by humans. As mentioned in previous work [d,u,v] and in our paper (Line 48 - 53), human feedback on chain-of-thought reasoning is challenging to collect. Thus, it is not realistic to annotate an adequate amount of reasoning steps to conduct SFT in practice. Second, it is not feasible to conduct SFT on the reasoning step directly generated by the KG, considering the heterogeneity between LLM reasoning and knowledge graph structures. Instead, our approach aligns the models on the generated reasoning steps (without relying on the reasoning steps collected from humans), by bridging the above heterogeneity between knowledge graphs and LLM reasoning forms through IPS policy estimation.

---

> ### Author Response · Authors · 2024-11-22
> **Rebuttal to Reviewer uYKA (Part 2/2)**
>
> **Response to Question 2,3,4,5,7,8,9**
>
> Q2: The offline policy evaluation performance without performing updated according to equation 4 is exactly the reported **Base** performance in Table 1 and Table 2.
>
> Q3 and Q8: The currently reported SFT performance is the performance of SFT + normal COT in Table 1, Table 2, and Table 3.
>
> Q4: As detailed in Section 4.1, we explain the calculation of policy evaluation in Equation 3.
>
> Q5: OCEAN and SFT are both fine-tuned based on the Base model.
>
> Q7: Since the index k is for each question q and generated answer y, the equation for the policy gradient should be subscribed by k, i.e., $\mu_{\phi}(y_{k,t}|q_{k},y_{k,<t})$. We will edit this accordingly to avoid confusion.
>
> Q9: Following the existing setting for knowledge-intensive and multi-hop question-answering (Line 293 - 310),
> we study the dual experimental settings where the LLM's prompt only contains the query (w/o ctx),
> and the relevant context of knowledge provided by the dataset is also included in the prompt (w/ ctx).
> We will better highlight such information in our experimental settings.
>
> [a] Levine, Sergey, et al. "Offline reinforcement learning: Tutorial, review, and perspectives on open problems." arXiv preprint arXiv:2005.01643 (2020).
>
> [b] Gilotte, Alexandre, et al. "Offline a/b testing for recommender systems." Proceedings of the Eleventh ACM International Conference on Web Search and Data Mining. 2018.
>
> [c] Jeunen, Olivier. "Revisiting offline evaluation for implicit-feedback recommender systems." Proceedings of the 13th ACM conference on recommender systems. 2019.
>
> [d] Joshi, Nitish, et al. "Improving Multi-Hop Reasoning in LLMs by Learning from Rich Human Feedback." Neuro-Symbolic Learning and Reasoning in the era of Large Language Models. 2023.
>
> [e] Nguyen, Minh-Vuong, et al. "Direct Evaluation of Chain-of-Thought in Multi-hop Reasoning with Knowledge Graphs." arXiv preprint arXiv:2402.11199 (2024).
>
> [f] Dudík, Miroslav, John Langford, and Lihong Li. "Doubly robust policy evaluation and learning." arXiv preprint arXiv:1103.4601 (2011).
>
> [g] Wu, Yike, et al. "CoTKR: Chain-of-Thought Enhanced Knowledge Rewriting for Complex Knowledge Graph Question Answering." Proceedings of the 2024 Conference on Empirical Methods in Natural Language Processing. 2024.
>
> [h] Sun, Jiashuo, et al. "Think-on-Graph: Deep and Responsible Reasoning of Large Language Model on Knowledge Graph." The Twelfth International Conference on Learning Representations.
>
> [i] Zhang, Yu, et al. "Question-guided Knowledge Graph Re-scoring and Injection for Knowledge Graph Question Answering." Findings of the Association for Computational Linguistics: EMNLP 2024. 2024.
>
> [j] Fu, Peng, et al. "Revisiting the Knowledge Injection Frameworks." Proceedings of the 2023 Conference on Empirical Methods in Natural Language Processing. 2023.
>
> [k] Zhang, Shenao, et al. "Self-Exploring Language Models: Active Preference Elicitation for Online Alignment." Automated Reinforcement Learning: Exploring Meta-Learning, AutoML, and LLMs.
>
> [l] Ding, Mucong, et al. "SAIL: Self-improving Efficient Online Alignment of Large Language Models." ICML 2024 Workshop on Theoretical Foundations of Foundation Models.
>
> [m] Wang, Jianing, et al. "Knowledge Prompting in Pre-trained Language Model for Natural Language Understanding." Proceedings of the 2022 Conference on Empirical Methods in Natural Language Processing. 2022.
>
> [n] Fang, Tianqing, et al. "Complex reasoning over logical queries on commonsense knowledge graphs." arXiv preprint arXiv:2403.07398 (2024).
>
> [o] Fu, Xiyan, and Anette Frank. "The Mystery of Compositional Generalization in Graph-based Generative Commonsense Reasoning." Findings of the Association for Computational Linguistics: EMNLP 2024. 2024.
>
> [p] Wang, Zhichun, and Xuan Chen. "DERA: Dense Entity Retrieval for Entity Alignment in Knowledge Graphs." arXiv preprint arXiv:2408.01154 (2024).
>
> [q] Liu, Anqi, Hao Liu, Anima Anandkumar, and Yisong Yue. "Triply robust off-policy evaluation." arXiv preprint arXiv:1911.05811 (2019).
>
> [r] Dasgupta, Sutanoy, et al. "Off-Policy Evaluation Using Information Borrowing and Context-Based Switching." arXiv preprint arXiv:2112.09865 (2021).
>
> [s] Farajtabar, Mehrdad, Yinlam Chow, and Mohammad Ghavamzadeh. "More robust doubly robust off-policy evaluation." \textit{International Conference on Machine Learning}. PMLR, 2018.
>
> [t] Shi, Chengchun, et al. "Deeply-debiased off-policy interval estimation." International conference on machine learning. PMLR, 2021.
>
> [u] Xu, Weiwen, et al. "Exploiting Reasoning Chains for Multi-hop Science Question Answering." Findings of the Association for Computational Linguistics: EMNLP 2021. 2021.
>
> [v] Zhao, Wenting, et al. "Hop, Union, Generate: Explainable Multi-hop Reasoning without Rationale Supervision." Proceedings of the 2023 Conference on Empirical Methods in Natural Language Processing. 2023.

---

> ### Comment · Reviewer_uYKA · 2024-11-24
>
> I would like to thank the reviewer for their response and would like to ask a few follow-up questions and clarification questions.
>
> **Follow-up to Question 6**: I assume this paper addresses the issues related to scaling human preference data by utilizing knowledge graph preferences instead. Specifically, you convert chains of triplets from the knowledge graph into chains of thoughts using a model f to verbalize them. This suggests that you have a computational oracle (i.e., a verbalizer) rather than a human oracle to query, correct? If this is the case, then you are not relying on human feedback for reasoning chains and can actually scale beyond the limitations associated with human preferences.
>
> Yes, papers [d, u, v] mentioned that collecting human preference on the chain of thoughts is challenging, which I agree with. However, all of those papers include SFT-related experiments on their respective chain of thought datasets. In particular, [d] discusses how fine-tuning on small datasets with rich human feedback can improve model performance, while [u] presents a supervised training baseline where they directly train using the reasoning chain.
>
> In the context of this paper, I agree that the heterogeneity between LLM reasoning and knowledge graph structures is a problem. However, you address this issue by using an LLM to verbalize each training triplet. This means the heterogeneity issue is resolved, and you now have chain of thoughts reasoning data that you can train on, or am I misunderstanding something?
>
> **Follow-up to Question 1**: The key difference between DM OPE and IPS OPE lies in how we handle the distribution mismatch. DM OPE ignores this mismatch and estimates the reward using the exploration policy, while IPS OPE addresses the mismatch and uses the ground truth reward derived from the exploration policy.
>
> In the context of your problem, what is the distinction between the ground truth reward from the exploration policy and the possibly estimating reward? Currently, the reward function is \log \pi_0, which means that, in this case, both the ground truth and the estimate reward are the same with respect to DM and IPS, or am I misunderstanding? If this is the case, having an ablation where you remove the importance ratio \lambda term is important for me to understand where the performance benefits are coming from.
>
> **Clarification Question 1**: Are you initializing the policy with SFT or the Base model when implementing your OCEAN framework?

---

> > ### Author Response · Authors · 2024-11-25
> >
> > Thanks for the reviewer's response. We address the concerns raised and provide answers to your comments accordingly.
> >
> > **Response to Follow-up to Question 6**
> >
> > As the reviewer suggested, we further conducted supervised fine-tuning on  the questions and answers, as well as verbalized knowledge graph trajectories. We present the comparison results of Llama-3 and Gemma-2.
> >
> > | Llama-3              | StrategyQA | ARC   | SciQA | CSQA-COT1000 | OpenBookQA |
> > |----------------------|------------|-------|-------|--------------|------------|
> > | Base                 | 59.53      | 79.93 | 57.10 | 69.50        | 58.80      |
> > | SFT                  | 51.82      | 61.87 | 35.80 | 49.00        | 63.20      |
> > | SFT w/ Verbalized KG | 49.05      | 53.85 | 38.20 | 63.50        | 55.20      |
> > | OCEAN                | 59.83      | 80.60 | 57.70 | 72.00        | 60.40      |
> >
> > | Gemma-2              | StrategyQA | ARC   | SciQA | CSQA-COT1000 | OpenBookQA |
> > |----------------------|------------|-------|-------|--------------|------------|
> > | Base                 | 60.99      | 65.89 | 47.10 | 63.50        | 51.80      |
> > | SFT                  | 51.97      | 18.06 | 51.50 | 15.00        | 7.40       |
> > | SFT w/ Verbalized KG | 35.66      | 37.46 | 39.60 | 50.50        | 42.6       |
> > | OCEAN                | 66.23      | 63.21 | 47.50 | 72.50        | 57.20      |
> >
> > We observe that for smaller LLMs (e.g., Gemma-2) when SFT on only questions and answers, the model generation could degenerate while training with verbalized knowledge graph trajectories can occasionally improve to maintain the model's chain-of-thought generation ability. However, due to the heterogeneity between knowledge graphs and LLM reasoning, direct training on the verbalized knowledge graphs could still lead to sub-optimal LLM policy. OCEAN is not directly trained on knowledge graph trajectories but evaluated based on the IPS method, aligned by the importance ratio, and implicitly optimized by the estimated policy values, which alleviates such problems.
> >
> > We will add such a baseline to our comparison results and discuss such intuitions.
> >
> >
> > **Response to Follow-up to Question 1**
> >
> > Thanks for the clarified suggestion. As the reviewer suggested, we have done the ablation by removing the importance ratio term and conducted policy optimization compared with OCEAN and the Base model.
> >
> > | Llama-3                    | StrategyQA | ARC   | SciQA | CSQA-COT1000 | OpenBookQA |
> > |----------------------------|------------|-------|-------|--------------|------------|
> > | Base                       | 59.53      | 79.93 | 57.10 | 69.50        | 58.80      |
> > | OCEAN                      | 59.83      | 80.60 | 57.70 | 72.00        | 60.40      |
> > | OCEAN w/o importance ratio | 40.32      | 76.59 | 55.00 | 71.00        | 57.80      |
> >
> > | Gemma-2                    | StrategyQA | ARC   | SciQA | CSQA-COT1000 | OpenBookQA |
> > |----------------------------|------------|-------|-------|--------------|------------|
> > | Base                       | 60.99      | 65.89 | 47.10 | 63.50        | 51.80      |
> > | OCEAN                      | 66.23      | 63.21 | 47.50 | 72.50        | 57.20      |
> > | OCEAN w/o importance ratio | 40.03      | 58.86 | 39.13 | 72.00        | 53.6       |
> >
> > As the reviewer pointed out "DM OPE ignores this mismatch and estimates the reward using the exploration policy",  we observe that without the importance ratio, the performance of the ablation of OCEAN decreases consistently, especially in the knowledge-intensive tasks (e.g., SciQA, ARC), where the distribution mismatch between the target policy and the logging policy can be larger.
> > IPS measures the importance ratio to handle the distribution mismatch between the target policy and the logging policy, which aligns with the practical challenges of the heterogeneity between knowledge graphs and LLM reasoning.
> >
> > We will add such discussions to our paper to better motivate our methodology design.
> >
> >
> > **Response to Clarification Question 1**
> > For a fair comparison, our implementations of OCEAN and SFT are both initialized from the same Base model for each LLM backbone.

---

> > > ### Comment · Reviewer_uYKA · 2024-11-25
> > >
> > > Thank you so much for the quick response. I have one final clarification question.
> > >
> > > **Clarification Question**: The only conceptual difference between "SFT w/ Verbalized Knowledge Graph" and "OCEAN w/o importance ratio" lies in how the policy is updated, correct? The "OCEAN w/o importance ratio" updates the policy using policy gradient with the reward being the log \pi_0. In contrast, "SFT w/ Verbalized Knowledge Graph" does not utilize a reward. Additionally, during evaluation every policy is evaluated using equation 3, right?

---

> > > > ### Author Response · Authors · 2024-11-25
> > > >
> > > > Thanks for the reviewer's response. We confirm that the conceptual difference between these two methods is as the reviewer described. In addition, during evaluation, every policy is evaluated using equation 3. We will add such clarification in our paper to clear any confusion.

---

> ### Comment · Reviewer_uYKA · 2024-11-25
>
> Thank you for your quick response. I apologize for the additional questions; I want to make sure I understand what is going on.
>
> **Clarification Question 1**: The only conceptual difference between the "DM OPS" baseline and the proposed KG-IPS OPE is the \lambda term being removed from handling the distribution mismatch. Does this mean that gradients with respect to both the "DM OPS" baseline and proposed KG-IPS are the same, right? So both policies should perform exactly the same with respect to all downstream tasks except for the value function estimation. The value function estimation should be different because they are different OPEs.
>
> **Clarification Question 2**: The results you are showing above for "SFT w/ Verbalized KG" are w/o ctx, but I am curious about the performance improvement of SFT w/ ctx because you trained SFT w/ ctx.
>
> **Clarification Question 3**: I am a bit confused regarding the statement that commonsense reasoning tasks do not require CoT. I am not very familiar with the commonsense reasoning literature, but the original CoT paper showed that CSQA performance improved with CoT [a]. Additionally, a recent Anthropic paper demonstrated that CoT helps with OpenBookQA [b].
>
> [a] Chain-of-Thought Prompting Elicits Reasoning in Large Language Models by Wei et al.
>
> [b] Measuring Faithfulness in Chain-of-Thought Reasoning by Chen et al.

---

> > ### Comment · Reviewer_uYKA · 2024-11-26
> >
> > Since today is the last day of the discussion period, I would appreciate any feedback regarding any of my clarification questions. This will help me understand how to adjust my score.

---

> > > ### Author Response · Authors · 2024-11-27
> > >
> > > Thank you for your questions.
> > >
> > > **Response to Clarification 1**
> > >
> > > The gradients for DM-OPS and KG-IPS are indeed different due to the weight term. Referring to Equation (3), each token is reweighted by $\lambda(v|s_t^{(i)})$, and the average reward is computed using these weights. This reweighting directly modifies the value function $V$. From Equation (4), each component of $\lambda(v|s_t^{(i)})$ contributes to the value function, and the gradient reflects this contribution for optimization. The learnable parameter weight introduces flexibility in scaling the reward function, impacting parameter $\theta$ updates. Since DM is not the focus of our paper, initially we did not discuss this. We would like to emphasize that our paper and contributions focus on proposing (i) an offline evaluation framework to directly measure the quality of LLM chain-of-thought generation, (ii) an instantiation of the policy optimization method, and (iii) the theoretical analysis of the proposed estimator. We greatly appreciate the reviewer's questions and we would include the discussions in the updated version.
> > >
> > >
> > > **Response to Clarification 2**
> > >
> > > Thanks for the suggestion. We also include the SFT w/ Verbalized KG with ctx experiments. We observe that SFT w/ Verbalized KG achieves better or comparable performance of the original SFT.
> > >
> > > | Llama-3              | StrategyQA |  SciQA |
> > > |----------------------|------------|------------|
> > > | Base                 | 77.73      | 83.10  |
> > > | SFT                  | 66.52       | 67.10  |
> > > | SFT w/ Verbalized KG |68.85 | 67.00  |
> > > | OCEAN                | 75.40      | 83.20 |
> > >
> > > | Gemma-2              | StrategyQA | SciQA  |
> > > |----------------------|------------|--------|
> > > | Base                 | 76.71      | 76.50  |
> > > | SFT                  | 64.77      |  79.80 |
> > > | SFT w/ Verbalized KG | 68.90 | 77.60 |
> > > | OCEAN                | 74.24      | 72.20   |
> > >
> > > **Response to Clarification 3**
> > >
> > > We acknowledge that CoT can generally improve the commonsense reasoning tasks. We made that statement (i.e., commonsense reasoning tasks do not require CoT) only in Section 5.4, by considering the specific context of the experiments in that section. In Section 5.4, our experiment is designed to evaluate OCEAN's generalizability in commonsense knowledge and reasoning. Thus, the evaluation metric in Section 5.4 does not necessarily consider CoT generation evaluation.
> > >
> > > Although the above evaluation in Section 5.4 does not consider CoT generation evaluation, we are aware of the importance of evaluating CoT generation. We started our experiment in Section 5 by thoroughly evaluating CoT generation in Section 5.2 and 5.3 (i.e., evaluating chain-of-thought alignment achieved by OCEAN). We also included comprehensive experiments in Section 5.3 and Section 5.2. That is, we apply our developed framework in the same way in Section 5.2, 5.3, and 5.4, while the metrics considered in Section 5.4 and Section 5.2/5.3 are different to focus on evaluating the performance from different perspectives. Thank you for the comment. We will improve the description accordingly to enhance clarity.

---

> > > > ### Comment · Reviewer_uYKA · 2024-11-29
> > > >
> > > > Thank you for responding to my questions. I will update my score.

---

> > > > > ### Author Response · Authors · 2024-12-03
> > > > >
> > > > > We sincerely thank you for your thoughtful follow-up feedback, which has been invaluable in improving and clarifying our work.

---

### Official Review · Reviewer_hg5c · 2024-11-04

**Soundness:** 3
**Presentation:** 3
**Contribution:** 3
**Rating:** 8
**Confidence:** 4

**Summary:**

This paper focuses on evaluating Chain-of-Thought (CoT) reasoning with guidance from a Knowledge Graph (KG). The authors propose training a KG model as a reference and using KG-IPS to assess the target language model's performance based on the reference model’s output. They conduct extensive experiments to support this approach.

**Strengths:**

Overall, I find the paper’s idea and methodology compelling. The provided theoretical analysis is nice and the experimental results can support their statements.

**Weaknesses:**

However, improvements in writing would enhance readability and comprehension.

Please use the full name when introducing KG-IPS for the first time.

What role does the base LLM policy $\pi_0$ play? Including a figure to illustrate the entire pipeline—showing the knowledge graph reference policy $\mu_\phi$, target policy $\pi_\theta$, and base LLM policy $\pi_0$—would improve clarity and help readers follow the workflow more easily.

Could you expand on Equation 3, especially explaining the term $T_i | c_t^{(I)}$ and the intuition behind defining this term after the third summation symbol?

What does the x-axis in Figures 1(b) and 1(c) represent? This clarification is essential to verify the conclusions in the final paragraph of Section 4.

Readers might also wonder about the performance of the reference models and how the architecture of these models impacts the evaluation of the target model.

Finally, how would the reference model perform if it included process-based reward signals rather than relying solely on binary outcome signals defined by answer accuracy? Is a binary signal sufficient, or would process-based feedback add value?

If the author can address my concern, I would like to raise my rating.

**Questions:**

See weakness.

---

> ### Author Response · Authors · 2024-11-22
> **Rebuttal to Reviewer hg5c**
>
> Thank you for your valuable feedback and the time you have spent reviewing our work. We address the concerns raised and provide answers to your questions accordingly.
>
>
> **Response to Weakness 1**
>
> Thanks for pointing this out. We will fix this and make sure all abbreviations used are defined in their appearance.
>
>
> **Response to Weakness 2**
>
> Thanks for the suggestion. We added an illustrative figure in Appendix D (i.e., Figure 4) to demonstrate the evaluation pipeline to better explain the workflow.
>
>
> **Response to Weakness 3**
>
> The intuition behind defining this term after the third summation symbol is inspired by the Inverse Propensity Score (IPS) , which is widely used in reinforcement learning for off-policy evaluation to estimate the expected reward of a target policy $\pi_\theta$ using data collected from a different behavior policy.
> Here, $c$ represents the entire reasoning path, and $e$ denotes each token within the path. The log-likelihood score of each token under the base policy $\pi_0$ is used as the reward function.
> The term before the reward serves as a weighting factor to adjust the mismatch between the target policy $\pi_\theta$ and the behavior policy.
>
> Due to the reason that not all tokens are included in the KG trajectories, the calculation is two-fold.
> 1. For entity tokens, the reward is derived under the knowledge graph preference policy $\mu_\phi$, which serves as the behavior policy for these tokens.
> 2. For non-entity tokens, the reward is based on their likelihood under the base language model policy $\pi_0$. This can also be considered as a regularization term in the reward function.
> We then compute the sum of rewards for all tokens and normalize it across $T$ iterations.
>
>
> **Response to Weakness 4**
>
> The x-axis in Figures 1(b) and 1(c) represent the indices of sampled relations and entities ordered by their respective frequencies of appearance in the trajectories.
>
> Figures 1(b) and 1(c) illustrate the distribution of sampled relations and entities. The x-axis represents the relations and entities, ordered by their respective frequencies of occurrence in the trajectories, from the most frequent to the least frequent. For instance, in Figure 1(b), the most frequent relation appears over 1,200 times, while the second most frequent relation occurs approximately 500 times. As mentioned in Lines 275–280, we observe that the distribution of relations is notably skewed, with a stronger emphasis on the most frequent relations.
>
>
> **Response to Weakness 5**
>
> Since verbalization of knowledge graph reasoning is not the major contribution of OCEAN,
> we follow the standard model architectures in previous works [a,b,c] for reference modeling.
> To further understand the performance of the reference models of different architectures,
> we train the reference model based on GPT-Base, GPT-Medium, and GPT-XL.
> With different reference models, we evaluate both the base LLMs and OCEAN of Llama-3, Gemma-2, Phi-3.5, and Mistral-0.2.
>
> | GPT-Base | Gemma-2 | Llama-3 | Mistral-0.2 | Phi-3.5 |
> |:------|------------:|------------:|-------------:|--------------:|
> | Base  | -12.71 | -13.79 |  -12.36 |   -13.01 |
> | OCEAN | -12.07 | -13.35 |  -11.63 |   -12.65 |
>
> | GPT-Medium | Gemma-2 | Llama-3 | Mistral-0.2 | Phi-3.5 |
> |:------|------------:|------------:|-------------:|--------------:|
> | Base  | -23.2  | -24.43 |  -24.26 |   -23.68 |
> | OCEAN | -21.72 | -23.4  |  -23.07 |   -22.58 |
>
> | GPT-XL | Gemma-2 | Llama-3 | Mistral-0.2 | Phi-3.5 |
> |:------|------------:|------------:|-------------:|--------------:|
> | Base  |  -9.24 | -12.18 |  -11.87 |   -10.23 |
> | OCEAN |  -7.41 | -10.41 |  -10.53 |    -9.74 |
>
> We observe that with different architectures used for reference models, although the estimation scale  $\hat{V}(\theta)$ slightly differs due to the log probability scales from the output space of each model, there is a consistent improvement in the alignment of OCEAN across different backbone LLMs.
>
> **Response to Weakness 6**
>
> The answer accuracy in Figure 1(a) is used as a metric for (Line 258 - 269) sampling quality control to filter out unreasonable trajectories, which are used for the development of the reference model.
> During policy evaluation and optimization, the reference model provides weighting (In Equation 3) for every reward signal on each reasoning step in the process, which formulates the calculation of the inverse propensity score.
>
>
> [a] Guan, Jian, et al. "A knowledge-enhanced pretraining model for commonsense story generation." Transactions of the Association for Computational Linguistics 8 (2020): 93-108.
>
> [b] Andrus, et al. "Enhanced story comprehension for large language models through dynamic document-based knowledge graphs." In Proceedings of the AAAI Conference on Artificial Intelligence, vol. 36, no. 10, pp. 10436-10444. 2022.
>
> [c] Khorashadizadeh, Hanieh, et al. "Research Trends for the Interplay between Large Language Models and Knowledge Graphs." arXiv preprint arXiv:2406.08223 (2024).

---

> > ### Comment · Reviewer_hg5c · 2024-11-25
> >
> > Could you clarify Lines 258–269 further? Specifically, it would be helpful to elaborate on the process of training the reward model, as the current explanation feels unclear to me.

---

> ### Author Response · Authors · 2024-11-25
>
> Thanks for the reviewer's comment.
>
> In Lines 258 - 269, we first randomly collect a set of trajectories (in Equation 2), which contain a list of triplets for each trajectory. Each triplet contains a subject entity, a relation, and an object entity. Secondly, to further measure the validity of the sampled knowledge graph trajectories in terms of serving as reasoning paths, we follow the conventional self-consistency measurement [1,2] to prompt the GPT-4 model to propose a question about the first token in the trajectory, whose answer should be the last token. Therefore, the verbalized knowledge graph trajectory can be used as the reasoning paths of the question-answering pair. Then, we prompt the GPT-4 model to evaluate that given the proposed question and the verbalized knowledge graph trajectory, how likely the GPT-4 model could accurately answer the question. Such self-consistency measurement results in Figure 1(a), where such QA accuracy probability is used as the reward function (in LIne 265) for each verbalized knowledge graph trajectory. Finally, in Line 274, the KG preference model is trained on the verbalized knowledge graph trajectories with the developed reward function.
>
> We hope our response addresses your concerns. Please let us know if there are any additional questions, and we are happy to discuss further.
>
> [1] Xuezhi Wang, Jason Wei, Dale Schuurmans, Quoc Le, Ed Chi, Sharan Narang, Aakanksha Chowdhery, and Denny Zhou. Self-consistency improves chain of thought reasoning in language models.
> arXiv preprint arXiv:2203.11171, 2022b.
>
> [2] Potsawee Manakul, Adian Liusie, and Mark JF Gales. Selfcheckgpt: Zero-resource black-box hallucination detection for generative large language models. arXiv preprint arXiv:2303.08896, 2023.

---

> > ### Comment · Reviewer_hg5c · 2024-11-25
> >
> > Therefore, as randomly select the number of entities so that the reward model can provide step-wise supervision, right?

---

> > > ### Author Response · Authors · 2024-11-25
> > >
> > > The reward function in Line 274 $R(h|c)$ is the QA accuracy probability on each verbalized knowledge graph trajectory so that the reward function $R(h|c)$ only provides sample-wise feedback on each trajectory for the training of the preference model $\mu_\phi$.
> > >
> > > In addition, since the preference model $\mu_\phi$ is trained with verbalized knowledge graph trajectories (containing the randomly selected the number of entities), it can provide step-wise feedback, which is used in our proposed KG-IPS estimator in Equation 3.

---

> > > > ### Comment · Reviewer_hg5c · 2024-11-25
> > > >
> > > > Thanks for your response to my question. I have raised my score.

---

> > > > > ### Author Response · Authors · 2024-11-25
> > > > >
> > > > > Thanks for your recognition of our work and your valuable time in reviewing our paper.

---

### Official Review · Reviewer_wXum · 2024-11-07

**Soundness:** 3
**Presentation:** 4
**Contribution:** 3
**Rating:** 8
**Confidence:** 3

**Summary:**

The paper proposes a novel technique for performing offline evaluation of the chain-of-thought capabilities of LLMs using trajectories generated on knowledge graphs. Specifically they propose a well thought out framework, coined OCEAN, which models CoT reasoning in language models as an MDP, where they leverage on-policy reinforcement learning methods to learn a knowledge graph policy which generates likelihood distributions for LLM reasoning paths. They then define and introduce the KG-IPS estimator to evaluate the policy, which they prove is unbiased while also providing a lower bound on its variance. They then discuss how this estimator can be used for policy optimization which would align LLMs to favor the reasoning paths generated in the knowledge graph. Finally, they conduct experiments to evaluate their approach in knowledge-intensive, multi-hop and common sense reasoning tasks compared to various baseline and supervised fine-tuned language models.

**Strengths:**

1. The paper considers a practical approach for the offline evaluation of LLMs, which is an important area of research given that the dominant approach of RLHF is highly labor intensive and expensive.
2. The notion of using knowledge graphs for LLM alignment is very interesting, and their approach consisting of verbalizing the KG trajectories is novel to the best of my knowledge.
3. The paper contains very detailed and thorough experiments to demonstrate their results empirically.
4. Theoretical proofs look sound.
5. The paper is very well-written and enjoyable to read.

**Weaknesses:**

1. One aspect of the methodology that was not clear is the case where there are multiple trajectories that lead to the correct answer, which I believe can often be the case in reasoning tasks solved by LLMs. How will this be represented in the knowledge graph preference policy? Additionally, does the current methodology hold the potential to penalize reasoning paths that are correct but just not identified in a knowledge graph?

2. In a similar vein, it is not clear that the knowledge graph used in this model (which seems to be Wikidata5M) is extensive enough to capture all possible reasoning paths given I believe it only contains entities and relations captured on wikipedia. I think it is imperative that the authors provide more information on the specific knowledge graph they are using and why it is sufficient to model reasoning queries given to LLMs.

3. It is not clear from the tables provided in Section 5 that the OCEAN model significantly outperforms the baseline. A more detailed discussion explaining this seems necessary.

**Questions:**

Some questions were raised above.

1. Why did you choose to 'verbalize' the trajectories generated on the knowledge graphs instead of inversely trying to reformat the LLM reasoning chain as a trajectory using entity and relationship linking? This approach seems initially more intuitive and less likely to cause errors?

---

> ### Author Response · Authors · 2024-11-22
> **Rebuttal to Reviewer wXum (Part 1/2)**
>
> Thank you for your valuable feedback and the time you have spent reviewing our work. We address the concerns raised and provide answers to your questions accordingly.
>
> **Response to Weakness 1**
>
> In our knowledge graph preference policy, we randomly sample reasoning trajectories to avoid potential bias on specific reasoning paths.
> However, in practice, exhausting every potential reasoning path is unrealistic and impractical.
> Thus, by leveraging the inverse propensity score (IPS), we adjust for selection bias by weighting observed data inversely proportional to the probability of selection [a,b,c].
> Theoretically, we demonstrate that the IPS estimator provides an unbiased estimate of the target policy by proving that the expected value of
> the IPS estimator equals the true expected reward under $\pi_\theta$.
>
>
> **Response to Weakness 2**
>
> Several prior works [d,e,f,g] have effectively utilized knowledge graphs to align LLM reasoning with more faithful multi-hop reasoning,
> underscoring the validity and effectiveness of our approach to evaluating and optimizing LLMs through knowledge graph-based methods.
> While we do not claim that Wikidata5M encompasses all possible reasoning paths, we selected it due to its recognized comprehensiveness [j,k,l].
> This choice allowed us to demonstrate the robustness and adaptability of our method.
> In general, our approach is adaptable and can be applied to any popular knowledge graph.
> Empirically, the evaluated tasks in Sections 5.2, 5.3, and 5.4 are not associated with Wikidata5M,
> further emphasizing the generality and versatility of our approach.
> By enhancing the faithfulness of reasoning, OCEAN contributes to improving LLMs’ general reasoning abilities across diverse tasks.
>
>
> **Response to Weakness 3**
>
> We would like to emphasize that our contributions are the offline evaluation framework to directly measure the quality of LLM chain-of-thought generation, a direct policy optimization method, and theoretical analysis of the proposed estimator.
> OCEAN achieves better multi-step reasoning alignment based on policy evaluation $\hat{V}(\theta)$, which demonstrates the effectiveness of improving on the CoT generation quality. We show that OCEAN consistently improves $\hat{V}(\theta)$ on multi-step reasoning generation in multi-hop question-answering tasks (in Table 1), and achieves improvements or comparable performance on knowledge-intensive tasks relying on multi-step reasoning (in Table 2).
>
> In addition, we also show comparative results on the question-answering accuracy in multiple tasks (in Table 1, Table 2, and Table 3). However, since OCEAN does not directly learn on any supervised labels regarding question-answering tasks, the performance improvements only come from the offline policy evaluation and optimization method. Even without supervised learning on question-answering tasks, OCEAN achieves the best performance on HotpotQA and Strategy QA w/o ctx, while maintaining comparable performance on MuSiQue for multi-hop reasoning tasks (Table 1). We also observe that the best-performed LLMs on ARC, PubMedQA w/o ctx, and SciQA, are optimized by OCEAN (Table 2). In commonsense reasoning tasks, OCEAN-optimized Gemma-2 and Mistral-0.2 achieve better average performance too (Table 3).
>
> **Response to Question 1**
>
> We chose to verbalize knowledge graph trajectories instead of reformatting LLM reasoning chains as trajectories due to the inherent heterogeneity between LLM reasoning forms and structured knowledge graphs. As highlighted in Line 54 - 65 of [h], accurate grounding of reasoning chains on specific knowledge graphs is restrictive and prone to errors. Thus, directly reformatting the LLM reasoning chain as a trajectory using entity and relationship linking can be hard to adequately capture the complexity of LLM-generated thought processes and introduces bias in feedback [i].
> By treating knowledge graphs as weak yet controllable reasoners, our approach aligns multi-step reasoning with multi-hop trajectories through inverse propensity scores (IPS).
> This ensures a more robust evaluation, leveraging the controllable structure of knowledge graphs without rigidly relying on exact reformulation or grounding.

---

> ### Author Response · Authors · 2024-11-22
> **Rebuttal to Reviewer wXum (Part 2/2)**
>
> [a] Dudík, Miroslav, John Langford, and Lihong Li. "Doubly robust policy evaluation and learning." arXiv preprint arXiv:1103.4601 (2011).
>
> [b] Thorsten Joachims, Adith Swaminathan, and Tobias Schnabel. Unbiased learning-to-rank with biased feedback. In Proceedings of the tenth ACM international conference on web search and data mining, pp. 781–789, 2017.
>
> [c] Edward L Ionides. Truncated importance sampling. Journal of Computational and Graphical Statistics, 17(2):295–311, 2008.
>
> [d] Wu, Yike, et al. "CoTKR: Chain-of-Thought Enhanced Knowledge Rewriting for Complex Knowledge Graph Question Answering." Proceedings of the 2024 Conference on Empirical Methods in Natural Language Processing. 2024.
>
> [e] Sun, Jiashuo, et al. "Think-on-Graph: Deep and Responsible Reasoning of Large Language Model on Knowledge Graph." The Twelfth International Conference on Learning Representations.
>
> [f] Zhang, Yu, et al. "Question-guided Knowledge Graph Re-scoring and Injection for Knowledge Graph Question Answering." Findings of the Association for Computational Linguistics: EMNLP 2024. 2024.
>
> [g] Fu, Peng, et al. "Revisiting the Knowledge Injection Frameworks." Proceedings of the 2023 Conference on Empirical Methods in Natural Language Processing. 2023.
>
> [h] Nguyen, Minh-Vuong, et al. "Direct Evaluation of Chain-of-Thought in Multi-hop Reasoning with Knowledge Graphs." arXiv preprint arXiv:2402.11199 (2024).
>
> [i] Dudík, Miroslav, John Langford, and Lihong Li. "Doubly robust policy evaluation and learning." arXiv preprint arXiv:1103.4601 (2011).
>
> [j] Wang, Xiaozhi, et al. "KEPLER: A unified model for knowledge embedding and pre-trained language representation." \textit{Transactions of the Association for Computational Linguistics} 9 (2021): 176-194.
>
> [k] Wang, Liang, et al. "SimKGC: Simple Contrastive Knowledge Graph Completion with Pre-trained Language Models." \textit{Proceedings of the 60th Annual Meeting of the Association for Computational Linguistics (Volume 1: Long Papers)}. 2022.
>
> [l] Qiao, Zile, et al. "Improving Knowledge Graph Completion with Generative Hard Negative Mining." \textit{Findings of the Association for Computational Linguistics: ACL 2023}. 2023.

---

> > ### Comment · Reviewer_wXum · 2024-11-25
> > **Score update**
> >
> > Thank you for the detailed clarifications. I'm happy to increase my evaluation score.

---

> > > ### Author Response · Authors · 2024-11-27
> > >
> > > We sincerely thank you for your thoughtful follow-up feedback, which has been invaluable in improving and clarifying our work.

---

### Official Review · Reviewer_QYsv · 2024-11-08

**Soundness:** 3
**Presentation:** 3
**Contribution:** 3
**Rating:** 6
**Confidence:** 3

**Summary:**

The paper explores a novel approach for enhancing the chain-of-thought (CoT) reasoning in large language models (LLMs) through an offline evaluation and optimization framework. The proposed OCEAN framework utilizes knowledge graphs (KGs) to provide feedback and improve alignment in CoT tasks by modeling reasoning paths as a Markov Decision Process (MDP). By employing a KG-based inverse propensity score (KG-IPS) estimator, the framework enables unbiased offline evaluation, bridging the gap between LLMs and structured KG reasoning to ensure consistent, accurate CoT reasoning without undermining LLMs' performance in general tasks.

**Strengths:**

1. The concept of aligning CoT reasoning with knowledge graphs via a KG-IPS estimator is well-motivated, particularly in addressing the challenges associated with human feedback limitations and online deployment costs. The offline nature of the OCEAN framework allows for feasible and efficient model alignment using KGs as a structured feedback source.

2. The paper provides a thorough theoretical foundation for the KG-IPS estimator, proving its unbiasedness and offering a variance lower bound. This adds robustness to the evaluation framework and offers a sound basis for its application in LLM alignment.

3. The authors conduct a wide range of experiments across multi-hop question answering, knowledge-intensive tasks, and commonsense reasoning. These diverse evaluations demonstrate OCEAN's ability to align reasoning paths effectively while maintaining or enhancing the generalization capabilities of the LLMs across domains.

**Weaknesses:**

1. While the OCEAN framework is well-conceived, the reliance on multiple components (MDP formulation, KG-IPS estimator, verbalized KG policy) might limit its accessibility for practitioners. Simplifying or modularizing the system, potentially with more explicit guidance for implementation, could improve its applicability.

2. Limited Scope in KG-Based Reasoning Tasks: Although the alignment with KGs enhances factual consistency, the paper does not deeply explore its limitations in more abstract CoT reasoning, where LLMs rely on conceptual, rather than fact-based, reasoning. This could be an area for future work, as abstract reasoning is a critical facet of human-like reasoning in LLMs.

3. The paper focuses on large-scale LLMs, which are often resource-intensive to train and evaluate. While the experiments are comprehensive, further validation on smaller LLMs could strengthen the claim of OCEAN's adaptability across various scales.

**Questions:**

1. Could you elaborate on the specific challenges encountered in balancing entity and non-entity token weights in the KG-IPS estimator? How sensitive is the estimator to variations in these weights across different types of reasoning tasks?

2. Since your framework relies on structured knowledge graphs, how does OCEAN perform on CoT reasoning tasks that are less fact-based and more abstract or conceptual? Have you considered extending the model to handle these types of reasoning, and what might be the challenges?

3. Given the framework's reliance on multiple components (e.g., MDP modeling, KG-IPS estimator), what is the additional computational overhead compared to baseline CoT methods? Could you provide some insights into how to make OCEAN more computationally efficient?

4. How well does the OCEAN framework adapt to different types of knowledge graphs (e.g., smaller or domain-specific KGs)? What, if any, modifications are necessary when using a KG that differs substantially in structure or domain from Wikidata5M?

5. In some tasks, you noted the possibility of knowledge conflicts when aligning LLM outputs with knowledge graphs. How does OCEAN address or mitigate conflicts between an LLM's parametric knowledge and KG-based feedback, especially in cases where both knowledge sources might offer valid, but differing, information?

6. The paper mentions using sub-Gaussian concentration inequalities to establish confidence intervals for the KG-IPS estimator. Could you explain why this choice was made and how these intervals compare to other possible statistical measures?

---

> ### Author Response · Authors · 2024-11-22
> **Rebuttal to Reviewer QYsv (Part 1/2)**
>
> Thank you for your valuable feedback and the time you have spent reviewing our work. We address the concerns raised and provide answers to your questions accordingly.
>
> **Response to Weakness 1**
>
> In reinforcement learning, it is common to model LLMs as MDPs [a,b,c], especially for chain-of-thought reasoning [d,e,f].
> In our framework, the MDP formulation is only used to define the task and does not impact the KG-IPS estimator, which is the core component for evaluating the chain-of-thought generation process.
> The verbalized KG policy, as part of the IPS estimator, serves as the behavior policy to align outputs with knowledge graph preferences.
>
> While direct on-policy optimization could theoretically be used, it is practically challenging due to the engineering effort required for step-by-step interactions with structured knowledge graphs.
> Our offline evaluation and optimization approach provides a simpler and more practical solution, reducing complexity while ensuring alignment with knowledge graph preferences.
>
>
> **Response to Weakness 2 and Question 2**
>
> Our method is not restricted to fact-based problems, as knowledge graphs often encompass abstract or conceptual knowledge. For instance, in Wikidata5M, which we use in our experiments, examples of trajectories involving abstract or conceptual knowledge include: (groups and organizations, subsystem, subclass of), (Taxonomy, Classification scheme, is a type of). In these trajectories, the third element in the tuple represents the relationship between the first two elements. By leveraging such trajectories, our method extends beyond fact-based knowledge.
> In addition, we evaluated OCEAN on commonsense question-answering tasks in Section 5.4, where we assessed its general reasoning ability.
>
>
> **Response to Weakness 3**
>
> We conducted experiments using Gemma-2 (2.8B), Phi-3.5-mini (3.8B), Llama-3 (8B), and Mistral-0.2 (7B), where multiple sizes of LLMs are assessed and they are widely used in previous works [g,h,i]. Besides, in our experiments, we found that smaller models, such as Gemma-2 and Phi-3.5-mini, exhibit limited ability to follow instructions effectively compared to larger models like Llama-3 and Mistral-0.2 [j,k]. Consequently, for even smaller LLMs, where instruction-following capabilities are inadequate, investigating issues related to factuality would lack practical relevance [l,m,n].
>
> **Response to Question 1**
>
> The primary challenge in balancing entity and non-entity token weights in the KG-IPS estimator lies in controlling the variance caused by large behavior discrepancies between the policies $\mu_\phi$ and $\pi_\theta$. The estimator avoids excessive variance accumulation by introducing separate weights for different tokens.
> From a theoretical perspective, the KG-IPS estimator is unbiased, ensuring accurate evaluation of the target policy $\pi_\theta$. Lemma 2 also shows that the estimator’s variance is bounded by $M^2/n$. This bound, combined with the confidence intervals $O(M \sqrt{\log(1 / \delta) / n})$, ensures robustness and stability, making the estimator reliable across different reasoning tasks.
>
> **Response to Question 3**
>
> Components such as the MDP modeling and the KG-IPS estimator are employed solely to facilitate the fine-tuning process and are not incorporated into the model as additional components.
> Since OCEAN is designed for offline policy evaluation and optimization, the KG-IPS estimator is only trained offline based on the reference model and used for LLMs' offline alignment.
> During the inference time, there is no computational overhead of OCEAN compared to the baseline CoT methods, as both approaches are prompted with the same prompt and require the same amount of computational resources.  Consequently, our method does not introduce any extra parameters or computational demands, as it focuses exclusively on implementing a fine-tuning mechanism without altering the original model architecture.
>
>
> **Response to Question 4**
>
> Since we leverage verbalized knowledge graph trajectories as offline logged data for policy evaluation and optimization, OCEAN does not rely on any specific graph structures or domain knowledge, which is explained in Line 60 - 65. We choose Wikidata5M as an instantiation by following [s,t,u]

---

> ### Author Response · Authors · 2024-11-22
> **Rebuttal to Reviewer QYsv (Part 2/2)**
>
> **Response to Question 5**
>
> Instead of previous works on knowledge editing or injection of LLMs that cater to specific downstream tasks, OCEAN is designed to enhance LLMs' faithfulness when generating chain-of-thought reasoning paths based on a given context.
>
> To achieve this, we developed KG-IPS, a method for offline evaluation and optimization that measures the alignment between the likelihood of LLM-generated reasoning and a knowledge graph.
> Thus, OCEAN does not rely on ground truth labels (Lines 317–320) or enforce alignment to any domain-specific knowledge.
>
> In addition, the evaluated tasks in Sections 5.2, 5.3, and 5.4 are not associated with Wikidata5M, underscoring the generality of our approach.
> By enhancing the faithfulness of reasoning, OCEAN contributes to improving LLMs’ general reasoning abilities across diverse tasks.
>
>
> **Response to Question 6**
>
> In the KG-IPS estimator, sub-Gaussian is commonly used to calculate the lower bound of the variance [o,p,q,r].
> Compared to other measures, such as asymptotic normal approximations or looser Markov inequalities, sub-Gaussian bounds are non-asymptotic and offer sharper confidence intervals in finite-sample settings.
> These properties are especially useful in offline reinforcement learning scenarios where the number of samples is limited and fixed, and large discrepancies between policies can lead to high variance.
> Sub-Gaussian inequalities help keep the estimator reliable and stable in such situations.
>
>
> [a]. Zhang, Zongmeng, et al. "Trustworthy Alignment of Retrieval-Augmented Large Language Models via Reinforcement Learning." arXiv preprint arXiv:2410.16843 (2024).
>
> [b]. Gholamian, Sina, and Domingo Huh. "Reinforcement Learning Problem Solving with Large Language Models." arXiv preprint arXiv:2404.18638 (2024).
>
> [c]. Hu, Bin, et al. "Enabling intelligent interactions between an agent and an LLM: A reinforcement learning approach." arXiv preprint arXiv:2306.03604 (2023).
>
> [d] Yang, Mengjiao Sherry, et al. "Chain of thought imitation with procedure cloning." \textit{Advances in Neural Information Processing Systems} 35 (2022): 36366-36381.
>
> [e] Jiao, Fangkai, et al. "Learning planning-based reasoning by trajectories collection and process reward synthesizing." \textit{arXiv preprint arXiv:2402.00658} (2024).
>
> [f] Wang, Chaojie, et al. "Q*: Improving multi-step reasoning for llms with deliberative planning." \textit{arXiv preprint arXiv:2406.14283} (2024).
>
> [g] Li, Mo, et al. "NeedleBench: Can LLMs Do Retrieval and Reasoning in 1 Million Context Window?." \textit{arXiv preprint arXiv:2407.11963} (2024).
>
> [h] Murthy, Rithesh, et al. "MobileAIBench: Benchmarking LLMs and LMMs for On-Device Use Cases." \textit{arXiv preprint arXiv:2406.10290} (2024).
>
> [i] Pramanick, Shraman, Rama Chellappa, and Subhashini Venugopalan. "SPIQA: A Dataset for Multimodal Question Answering on Scientific Papers." \textit{The Thirty-eight Conference on Neural Information Processing Systems Datasets and Benchmarks Track}.
>
> [j] Lou, Renze, Kai Zhang, and Wenpeng Yin. "Large Language Model Instruction Following: A Survey of Progresses and Challenges." \textit{Computational Linguistics} (2024): 1-10.
>
> [k] Chung, Hyung Won, Le Hou, Shayne Longpre, Barret Zoph, Yi Tay, William Fedus, Yunxuan Li et al. "Scaling instruction-finetuned language models." \textit{Journal of Machine Learning Research} 25, no. 70 (2024): 1-53.
>
> [l] Lee, Nayeon, et al. "Factuality enhanced language models for open-ended text generation." \textit{Advances in Neural Information Processing Systems} 35 (2022): 34586-34599.
>
> [m] Guan, Jian, et al. "Language Models Hallucinate, but May Excel at Fact Verification." NAACL. 2024.
>
> [n] Tian, Katherine, et al. "Fine-Tuning Language Models for Factuality." \textit{The Twelfth International Conference on Learning Representations}.
>
> [o]. Strehl, Alex, et al. "Learning from logged implicit exploration data." Advances in neural information processing systems 23 (2010).
>
> [p]. Alizadeh, Shima, et al. "Pessimistic off-policy multi-objective optimization." International Conference on Artificial Intelligence and Statistics. PMLR, 2024.
>
> [q]. Agarwal, Rishabh, Dale Schuurmans, and Mohammad Norouzi. "An optimistic perspective on offline reinforcement learning." International conference on machine learning. PMLR, 2020.
>
> [r]. Ma, Jiaqi, et al. "Off-policy learning in two-stage recommender systems." Proceedings of The Web Conference 2020. 2020.
>
> [s] Chepurova, Alla, et al. "Better Together: Enhancing Generative Knowledge Graph Completion with Language Models and Neighborhood Information." \textit{The 2023 Conference on Empirical Methods in Natural Language Processing}.
>
> [t] Yu, Donghan, and Yiming Yang. "Retrieval-enhanced generative model for large-scale knowledge graph completion." SIGIR, 2023.
>
> [u] Liu, Ben, et al. "UniLP: Unified Topology-aware Generative Framework for Link Prediction in Knowledge Graph." \textit{Proceedings of the ACM on Web Conference 2024}. 2024.

---

### Meta-Review · Area_Chair_B86s · 2024-12-20

**Metareview:**

This paper proposes a way to do an offline evaluation of the chain-of-thought capabilities of LLM using a knowledge graph and an inverse-propensity-based metric. Reviewers are generally positive on this one and found the idea of aligning knowledge graph reasoning and chain-of-thought to be interesting. Reviewers also appreciated the theoretical claims behind the proposed IPS approach. The main concerns were whether knowledge-graphs can capture abstract reasoning (reviewer QYsv), key concepts not explained (reviewer 3w4e), and whether experiments are convincing (reviewer wXum).

Personally, I found the paper hard to parse and agree with reviewer 3w4e's initial assessment. To begin with, the data generation process isn't clearly explained. Reading the unbiasedness proof of KG-ILP, it appears that certain tokens are generated by the knowledge graph policy $\mu_\phi$ and others by the LLM policy $\pi_0$, however, exactly how isn't very clear.  The notations don't make sense in Line 867 and Line 881 which show e being sampled twice. Further, why is the probability of token $v \in c^{(i)}_t$ in Equation 3 not dependent on the previous tokens in $c^{(i)}_t$. It isn't explained on line 211 why log-likelihood of a token under the base policy is used as the reward. Overall, the structure of the unbiasedness and variance is a standard analysis for IPS metrics. The result for variance shows, if anything, that it can be high if $M$ is high which is undesirable, although high variance is a general problem with IPS metrics. One way to fix this would be to do doubly-robust estimation using a direct method along with IPS (e.g., see Dudik et al., 2011 which is cited).

Overall, this one is hard to place since I think the paper isn't well presented despite generally positive sentiment from the reviewers. I think the biggest reason for acceptance is the innovativeness of the idea and positive results. For this reason, I am recommending a weak accept but wouldn't mind if the paper is not accepted this time. If the paper does get accepted, I would request authors to include a figure or explain in detail how the offline data is generated and motivate why different decisions are made (e.g., why is reward function given by log-likelihood under $\pi_0$).

**Additional Comments On Reviewer Discussion:**

Reviewers were generally happy. The main concerns were:

1. Whether knowledge graphs have information for helping with abstract reasonings or can they only help with factuality. The authors gave examples of abstract reasoning which shows a scope beyond factuality, although I can imagine subtle reasonings that can be hard to capture in a knowledge graph.

2. Questions about why not do online learning were raised by reviewer uYKA. The author's response was that getting human feedback can be expensive which is reasonable.

3. There were other questions about clarity and experiments. Of these, the most concerning for me was clarity which is my main concern here.

Overall, I feel there is an interesting and novel idea here but the paper can do a lot more in terms of better presentation fo the key ideas. Using a main figure for outline or pseudocode can go a long way. I would urge authors to improve the presentation in future revisions.

---

### Decision · Program_Chairs · 2025-01-22

Accept (Poster)